# Aerosol first indirect effect of African smoke at cloud base of marine cumulus clouds over Ascension Island, south Atlantic Ocean

Martin de Graaf[1], Karolina Sarna[2], Jessica Brown[3], Elma Tenner[2], Manon Schenkels[4], and David P. Donovan[1]

[1]Royal Netherlands Meteorological Institute (KNMI) R&D Satellite Observations, De Bilt, The Netherlands
[2]Geosciences & Remote Sensing Department, Delft University of Technology (TUD), Delft, The Netherlands
[3]Wageningen University, Meteorology and Air Quality Department, Wageningen, The Netherlands
[4]Utrecht University, Institute for Marine and Atmospheric Research, Utrecht, The Netherlands

**Correspondence:** M. de Graaf, martin.de.graaf@knmi.nl

**Abstract.**

The interactions between aerosols and clouds are among the least understood climatic processes and were studied over Ascension Island. A ground-based UV-polarisation lidar was deployed on Ascension, which is located in the stratocumulus to cumulus transition zone of the southeast Atlantic Ocean, to infer cloud droplet sizes and droplet number density near the cloud base of marine boundary layer cumulus clouds. The aerosol-cloud interaction (ACI) due to the presence of smoke from the African continent was determined during the monsoonal dry season. In September 2016, a cloud droplet number density $ACI_N$ of $0.3 \pm 0.21$ cm$^{-3}$ was found, and a cloud effective radius $ACI_r$ of $-0.18 \pm 0.06$ $\mu$m, due to the presence of smoke in and under the clouds. Smaller droplets near the cloud base makes them more susceptible to evaporation and smoke in the marine boundary layer over the southeast Atlantic Ocean will likely accelerate the stratocumulus to cumulus transition. The lidar retrievals were tested against more traditional radar-radiometer measurements and shown to robust and at least as accurate as the lidar-radiometer measurements. The lidar estimates of cloud effective radius are consistent with previous studies of cloud base droplet sizes. The lidar has the large advantage of retrieving both cloud and aerosol properties using a single instrument.

*Copyright statement.* TEXT

## 1 Introduction

The importance of low level marine boundary layer (MBL) clouds for the Earth's radiative energy has long been recognized. Their high albedo (30–40%) over a dark ocean reduces the flux of solar radiation into the ocean, while contributing only slightly to the downward thermal radiation, due to their low altitude (and thus high temperature) inside the MBL (Albrecht et al., 1988). An estimated 4% increase in MBL cloud cover could offset the warming due to a doubling of $CO_2$ (Randall et al., 1984). Aerosols are expected to modulate the low level cloud cover through an aerosol-induced reduction of precipitation (Albrecht, 1989; Ackerman et al., 2000) or change the cloud shortwave albedo through an increase in cloud condensation

nuclei (CCN) (Twomey, 1974, 1977). The increase in CCN could lead to an increase of cloud droplet number density and a decrease of cloud droplet size, provided that the moisture content is constant. This effect is known as the first aerosol indirect effect. Additionally, the absorption of shortwave radiation by aerosols will locally heat the atmosphere and may modulate cloud properties by enhancing evaporation (e.g. Wang et al., 2003; Xue and Feingold, 2006) or changes in the thermodynamic
stability.

In the subtropics, extensive stratocumulus cloud decks form over the pool of cold water created by upwelling ocean currents west of the continents. The descending branch of the Hadley circulation in the subtropics creates a strong temperature inversion at the top of the MBL, which the stratocumulus decks are generally unable to penetrate. The stratocumulus decks are maintained by radiative cooling at the top of the MBL. This creates a moist, well-mixed layer over a cold ocean surface. Trade
winds transport this system northwest along a gradient in sea surface temperature towards the warmer equator, and a transition to cumulus clouds is observed, driven by increased convection from the warmer underlying surface. When the cumulus clouds penetrate the inversion and entrain warm, dry air from the free troposphere, the stratocumulus cloud deck breaks up and gradually dissipates (e.g. Paluch and Lenschow, 1991; Bretherton and Wyant, 1997; Wyant et al., 1997). This generally accepted thermodynamic theory of stratocumulus to cumulus transition (SCT) observed in the subtropical oceans, is complicated when
precipitation (Yamaguchi et al., 2017) or the presence of aerosols are taken into account (Wang et al., 2003).

Aerosols have several reported competing effects on the SCT duration, depending on the vertical and horizontal distribution of the aerosols, age and composition of the aerosols, etc. Over Africa, smoke is injected into the atmosphere during the dry season of the monsoon, which is July–October in southern Africa (e.g. de Graaf et al., 2010; Zuidema et al., 2018), and transported over the southeast Atlantic Ocean (SEAO) in the free troposphere under influence of the anticyclic circulation over
Africa (Garstang et al., 1996; Swap et al., 1996) and the Southern African Easterly Jet (Adebiyi and Zuidema, 2016). Close to the continent, the smoke in the free troposphere is found well above the temperature inversion, separated from the cloud top, while further out over the ocean it is more often mixed with the cloud-top after several days of transport following the subsiding large-scale circulation. Therefore, near the continent, the smoke in the free troposphere was found to delay the SCT, by strengthening the temperature inversion at the top of the MBL during the day, when smoke absorbs solar radiation and heats
the atmosphere locally. The stronger inversion results in thicker stratocumulus (Johnson et al., 2004; Yamaguchi et al., 2015). Further from the continent, smoke was found entrained into the cloud layer (Painemal et al., 2014; Rajapakshe et al., 2017), changing the cloud droplet number density (Diamond et al., 2018) and decreasing the low-level cloud cover (Ajoku et al., 2021), due to a weakening of the temperature inversion and evaporation of smaller cloud droplets (Johnson et al., 2004; Xue and Feingold, 2006; Zhou et al., 2017).

For precipitating clouds the effects are much more complicated (e.g. Zhou et al., 2017) and therefore precipitating clouds are not considered here.

Inside the MBL aerosols are typically mixtures of sea salt and smoke from the African continent during the biomass burning season. The composition of the aerosol mixtures changes during its residence inside the MBL, due to processing inside clouds, interaction with air and absorption of sunlight (Dang et al., 2022). The shortwave radiation absorption by smoke during the
55  day changes the diurnal thermodynamics of the MBL (Zhang and Zuidema, 2019).

In this paper, a method is explored to study aerosol-cloud interactions of smoke around the base of the clouds around Ascension Island. Here, we focus on the base of low-level broken cloud deck in the SEAO following the metrics specified in McComiskey et al. (2009) for changes in cloud droplet effective radius $R_{\text{eff}}$ and cloud droplet number density, as a function of changes in aerosol optical thickness $\tau_{\text{aer}}$ or aerosol extinction, as derived using one specific instrument, a UV-polarisation lidar.

Such an instrument was located on Ascension Island during one month in 2016 and one month in 2017 during the dry season in Africa. The UV-lidar was part of the measurement campaign CLARIFY-2017 (Haywood et al., 2021), partnering with several ground-based and aircraft campaigns described in section 2. The measurements in 2017 were affected by alignment problems, which resulted in a lower Signal-to-Noise Ratio (SNR) compared to the 2016 measurements. Therefore, the measurements from 2016 are used mainly to show the aerosol-cloud interactions (Sect. 3). The consistency of the lidar measurements was investigated by comparing with the abundant additional campaign measurements both in 2016 and 2017, described in Sect. 4.

Both aerosol properties from the aerosol layers and cloud properties from the cloud deck were derived from the lidar data, using a technique to infer cloud parameters based on polarisation change due to multiple scattering near the cloud base (Donovan et al., 2015). In this set-up, only one instrument is needed to study the impact of aerosols on cloud albedo by relating the aerosol number density to the cloud droplet number density (Sarna and Russchenberg, 2016). The details of the retrievals of the aerosol and cloud properties are described in Appendix A. The lidar beam will not penetrate deep into the cloud layer due to the large scattering cross section of the cloud droplets in the UV. Therefore, the cloud measurement results are valid near the cloud base. In this study we relate the cloud properties to an altitude of of 100 m above the cloud base height, see section A4.

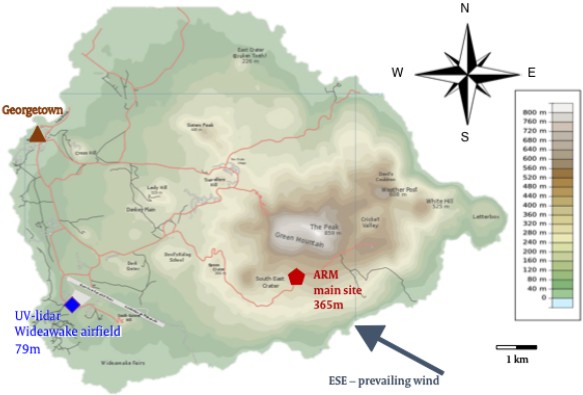

**Figure 1.** Map of Ascension Island, showing the topography and the location of the UV-lidar on Wideawake airfield and the ARM main site. The distance between the sites is 6.3 km. Georgetown is the island's main settlement.

## 2 Measurement campaign

From 3–29 September 2016 and from 15 August to 9 September 2017 the KNMI UV-polarisation lidar, normally located in Cabauw, The Netherlands, was relocated to Ascension Island, a remote volcanic island in the tropical Atlantic Ocean (8°S,14°W). Ascension Island is located 1600 km from the African coast and 2250 km from the Brazilian coast. Its climate is a tropical desert, with temperatures ranging from 22 to 31°C and a low annual rainfall at an average of 142 mm (Dorman and Bourke, 1981), the peak rainfall occurring in April. Ascension Island lies at the terminating stage of the SEAO SCT, with

clouds capping the boundary layer at an altitude of around 1–2 km. The prevailing trade winds in the boundary layer are from the south east (Kim et al., 2003) and mostly invariant. Above the boundary layer (> 1200 m above sea level) the wind is coming from the equatorial regions and frequently loaded with suspended particles like smoke from African vegetation fires or desert dust (Swap et al., 1996; Miller et al., 2021).

The Ascension Island Initiative (ASCII) was aimed at identifying microphysical properties of marine low level clouds in

the presence of aerosols (Brown, 2016; Tenner, 2017; Schenkels, 2018). During the same time various other measurements campaigns were operated on and around Ascension, providing a myriad of complementary measurements. The ground-based campaign LASIC (Zuidema et al., 2016) operated a fully-equipped ARM Research Facility during 2016 and 2017, while airborne measurements were provided during 2017 by CLARIFY-2017 (Haywood et al., 2021), and in 2016, 2017, and 2018 by ORACLES (Redemann et al., 2020). On the African continent, in-situ and airborne measurements of the smoke near the source

were provided by the AEROCLO-sA campaign in Namibia (Formenti et al., 2019).

Figure 1 shows the main locations of the instruments used in this paper during the campaigns. The UV-lidar was located on the southwest side of the island throughout the 2016 and 2017 campaigns on Wideawake airfield, at 79 m above sea level. For, all of 2016 and 2017, the ARM research facility was located on the south slope of Green Mountain, at 859 m the highest

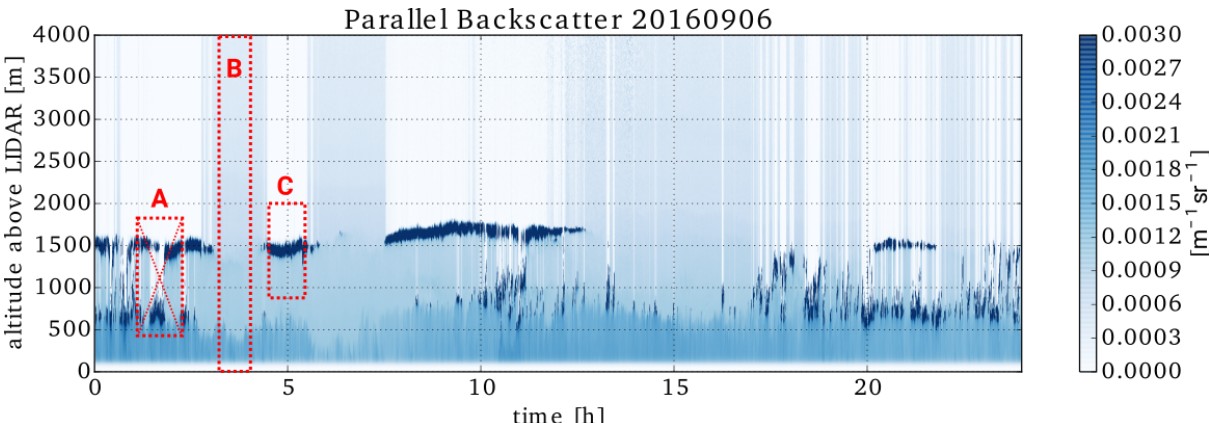

**Figure 2.** The parallel attenuated backscatter from the lidar on 6 Sept. 2016. The red boxes show examples of selected data: A) a cloud with varying cloud base and double cloud layers, not appropriate for analysis, B) appropriate selection of clear sky, C) appropriate selection of a cloud.

peak of the volcanic island. This location ensured that pristine oceanic air would be sampled. This air was transported in the
prevailing wind direction, which is east-southeast of the site. The ARM research facility was located at 365 m altitude and
about 6.3 km from Wideawake airfield. Radiosondes were launched from the airfield four times daily.

## 2.1 Lidar measurements

Lidar measurements have a long history of retrieving aerosol extinction and backscatter profiles in clear sky scenes (e.g.
Pappalardo et al., 2014). In aerosol conditions, the lidar signal is determined by single-scattering events. In clouds, multiple
scattering must be considered. The occurrence of multiple-scattering also has implication for the polarization state of the lidar
signal. Since cloud droplets are spherical, under single-scattering conditions, the lidar return signal retains its polarization
state. In clouds, multiple-scattering becomes more and more important as the beam penetrates from cloud base and the lidar
beam becomes increasingly depolarised. On Ascension Island, lidar measurements were performed to study both aerosol and
cloud properties, using a commercial Leosphere ALS-450 lidar operating at 355 nm, with separate parallel and perpendicular
channels. The data were acquired with a vertical resolution of 15 m and a temporal resolution of about 30 s. The field of view
of the lidar was found to be between 0.5 and 1.5 mrad. The retrieval error in 2016 was 19.75% and in 2017 39.05%, due to
the calibration, retrieval and measurement errors. In 2017, instrument internal misalignment (likely incurred during transport)
resulted in a lower SNR and uncertain calibration. Therefore, 2016 data are used in this paper, except where noted. The lidar
was operational 24 h per day for almost the entire period of the campaign, except from 24 to 27 Sept. 2016, due to power cuts
on the airfield. Details about the calibration and the campaign can be found in Brown (2016); Tenner (2017) and Schenkels
(2018).

An example of the type of both cloudy and clear-sky observation selected for analysis is presented in Fig 2. The skies over
Ascension are typically defined by broken low level warm clouds interspersed with clear spells. The lidar measurements were
used to estimate the aerosol and cloud properties during various circumstances, detailed below. Due to the strong background
light from the overhead sun, the ability to observe aerosols was much better at night or when no clouds were present.

## 2.2 Aerosol Optical Thickness

Using cloud-free lidar observations, the daily averaged AOT retrieved from the lidar during the 2016 campaign is shown in
Fig. 3, and compared to AERosol RObotic NETwork (AERONET) measurements from the station located on Ascension Island
at the ARM main site. AERONET offers quality-assured, cloud-screened automated direct sun measurements from ground-
based, sun-tracking sunphotometers every 15 minutes at 8 wavelengths (Holben et al., 1998). The measurements at 340 nm were
used here. The AERONET AOT data at this wavelength have an uncertainty of 0.021, due to atmospheric pressure variations
assuming a 3% maximum deviation from the mean surface pressure (Eck et al., 1999). The uncertainty of the lidar retrieval,
taking into account the systematic error arising from the definition of the extinction-to-backscatter ratios and the random error
due to the definition of the normalisation height, was estimated to be about 11% (Schenkels, 2018).

Daily averaged retrievals were compared for cloud-free periods for each instrument. Since the instruments were not at the
same position, the cloud-free periods can differ. However, the AOT distribution is assumed to be spatially consistent on the

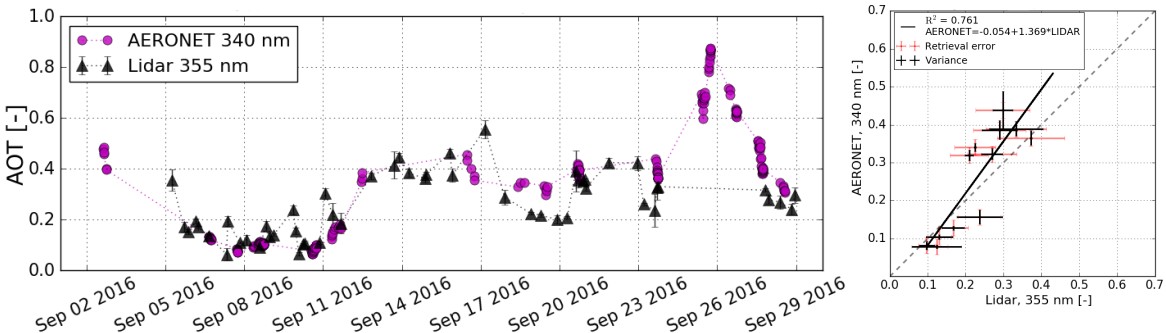

**Figure 3.** Aerosol Optical Thickness retrievals from AERONET at 340 nm compared to the retrieval from the UV-lidar at 355 nm. The left panel shows the daily averaged AERONET AOT (purple dots) during the 2016 campaign and the AOT from the lidar (black triangles) with black error bars showing the standard deviation. The retrieval uncertainties were 0.021 for AERONET and 11% for the lidar data. The right panel shows a scatter plot of the measurements on the left, with black bars showing the variances and red bars showing the retrieval errors. Pearson's correlation coefficient was 0.761. The dashed line the 1:1 line and the black full line a linear least-squares fit with slope 1.369 and offset -0.054.

spatial scale of around 6 km. The comparison between the AERONET and lidar retrieved AOT is good, with a correlation coefficient of 0.76.

The daily averaged AOT measurements show low aerosol conditions during the beginning of the campaign until 11 Sept. 2016, and increasing values until 17 Sept. 2016. After 17 Sept. the values decrease, but not to the same very low values as in the beginning of the month, and then again higher values towards the end. On 25 and 26 Sept. 2016, AERONET shows AOT values up to about 0.9, but unfortunately the lidar was not operational on those days. These values are consistent with 500 nm AERONET results, shown by Zuidema et al. (2018). AOT at 500 nm peaked in August 2016 and returned to low background values in the beginning of September 2016, as does the AOT at 340 nm. The increase in AOT over Ascension from 14–17 September and 23–26 September 2016 is consistent with the increase in strength of the Southern African Easterly Jet, which develops from weak in the beginning of Sept. 2016 to strong at the end of the month (Ryoo et al., 2022). This promoted the advection of black carbon (BC) from the African continent over the SEAO, suggesting that the AOT over Ascension Island increased due to the advection of smoke from Africa. This was also checked by inspection of daily backscatter trajectories, showing advection of air in the free troposphere directly from the east during the days with increased AOT (e.g. 13–17 Sept. and 23–26 Sept.), but not during low AOT episodes (e.g. 6-10 Sept. 2016). A few example backtrajectories during different episodes of the campaign are shown in Fig. 4.

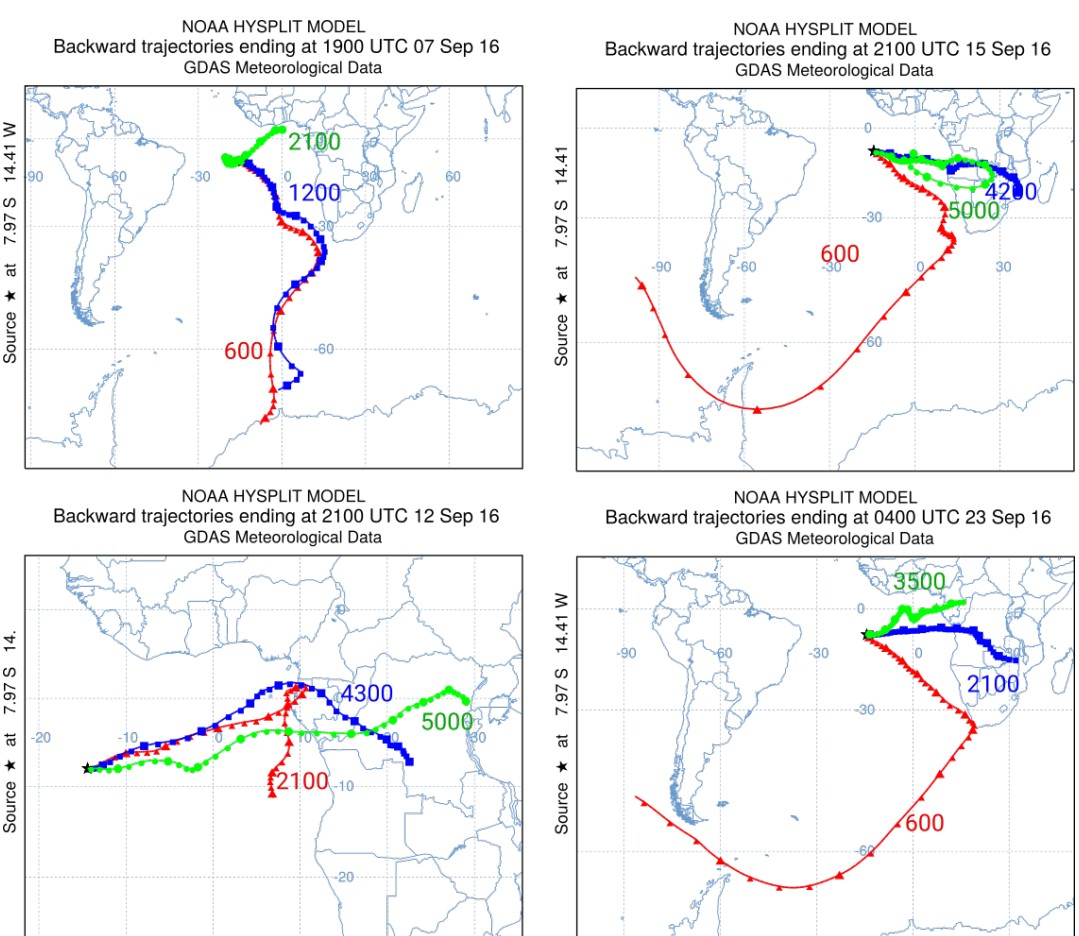

**Figure 4.** Example backtrajectories during the 2016 ASCII measurement campaign on 7,12,15 and 23 Sept. 2016. All trajectories were run for 240 h and ended over Ascension Island at different heights, as shown by the number (in m) altitude. The figures show the stable MBL east-southeast flow, and the advection of air from the African continent except on 7 Sept. 2016.

## 3  Aerosol-Cloud Interactions

Aerosol-cloud interactions were determined from the lidar measurements using the 2016 data only. In 2017, alignment issues resulted in a lower SNR and large uncertainties, and these data were discarded for the analyses in this section. Three approaches are presented. First, a simple comparison of days of low and high aerosol concentration is made, showing the change in cloud parameters. Next, the aerosol indirect effect was determined following the metrics developed in Feingold et al. (2001) and McComiskey et al. (2009): The aerosol-cloud interaction (ACI) is quantified, for a constant ambient relative humidity, by a the change in cloud parameters due to a change in the number of available condensation nuclei. For the cloud effective radius

$$\text{ACI}_r = -\frac{d\ln R_{\text{eff}}}{d\ln A}, \tag{1}$$

and for the cloud droplet number density

$$\text{ACI}_N = -\frac{d\ln N_d}{d\ln A}. \tag{2}$$

In these equations $A$ is the aerosol proxy, which should represent the aerosol abundance, and can be aerosol extinction, aerosol optical thickness or another aerosol quantity.

This approach was applied in two ways. First, by using the daily averaged AOT around the cloud base and comparing it to 155 the cloud parameters, which are also determined around cloud base (since the lidar does not penetrate deep into the cloud). Secondly, by determining the aerosol abundance below the cloud, using the lidar derived aerosol extinction profile below the clouds. Hence, in the first method the aerosol proxy is determined during cloud free spells, while in the second method the aerosol proxy is determined during cloudy spells, i.e. collocated in time with the cloud parameter retrievals. Aerosol optical thickness was retrieved using the classical Klett-Fernald two-mode method, i.e. applying Eqs. (A8) and (A9) to clear sky 160 measurements and cloud droplet number density and effective radius was retrieved by applying Eqs. (A12) and (A13) to measurements during cloudy periods.

### 3.1 Classification

A first coarse indication of the change in cloud properties can be obtained from a comparison of periods with a high aerosol loading over Ascension Island, compared to periods with low aerosol loading, assuming everything else will be the same. A 165 classification of the 2016 measurements was made after defining periods of clear sky and cloudy periods for each day with broken clouds.

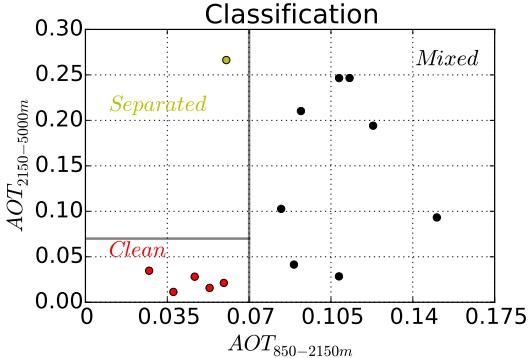

**Figure 5.** Classification of the average clear-sky AOT during broken cloud days, at two levels: from 850 to 2150 m, assumed to be at cloud level, and from 2150 to 5000 m, which is assumed to be in the free troposphere above the clouds.

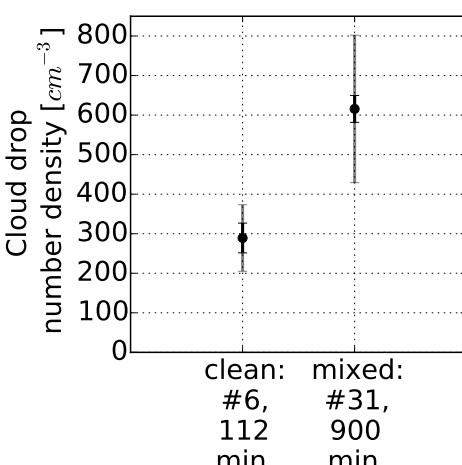 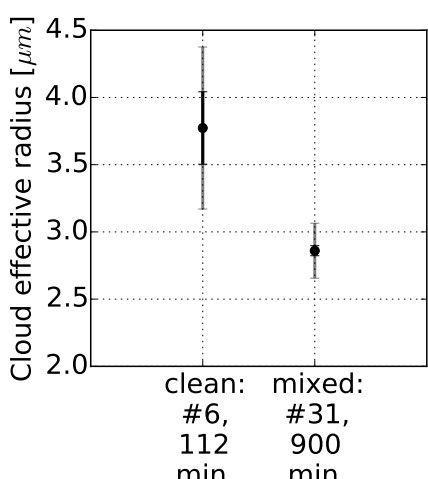

**Figure 6.** The mean value of the cloud droplet number density $N_d$ (left) and the cloud effective radius at reference height $R_{\text{eff}}^{100}$ (right) for the 'clean' and 'mixed' cases. The black error bar represents the standard deviation, the grey bars represent the sample standard deviation.

A classification was made of days when aerosols were expected to mix with the clouds and days when the aerosol loading was particularly low. Figure 5 explains the logic: two layers were discriminated, one from 850 to 2150 altitude, which was assumed to be the altitude of the clouds, and from 2150 to 5000 m, which was defined as the free troposphere. If the AOT in both layers was low (below 0.07 was chosen), the day was assigned the label 'clean', if the AOT in the layer between 850 and 2150 m was high (higher than 0.07), the days was assigned the label 'mixed'. If the AOT was high only in the free troposphere, the day was labeled 'separated' and not considered, which happened in one case. The average aerosol optical thickness was determined during the cloud free periods (6 in total), and the average cloud properties were determined during the cloudy periods (31 in total).

Using this crude selection of cases resulted in a clear difference in the average cloud properties between the different days, as shown in Fig. 6. The cloud droplet number density $N_d$ was $294\pm91$ cm$^3$ during all 'clean' days, doubling to over $611\pm191$ cm$^3$ during the 'mixed' days. Conversely, $R_{\text{eff}}^{100}$ was reduced from $3.81\pm0.6$ $\mu$m to $2.85\pm0.2$ $\mu$m. This suggests a change to smaller more numerous cloud particles with the availability of a larger number of cloud condensation nuclei. However, the assumption that the humidity does not change cannot be guaranteed with such an approach.

## 3.2 Aerosol-cloud interactions around cloud base

Next, the ACI was computed using AOT from the daily average extinction profile as before, but now averaged from 300 m below the cloud base until 1000 m above the cloud base. This level was chosen to isolate the MBL aerosol impact on cloud droplets near the cloud base, the region that the lidar is sensitive to. For each cloudy period the cloud properties were determined as before and used in Eqs. (1) and (2) to quantify the indirect effect. The results are shown in Fig. 7.

A linear fit was drawn through the points, weighted by the associated standard deviation, showing the indirect effect: $0.3 \pm 0.21$ cm$^{-3}$ for the cloud droplet number density and $-0.18 \pm0.06$ $\mu$m for the cloud effective radius. The indirect effect of

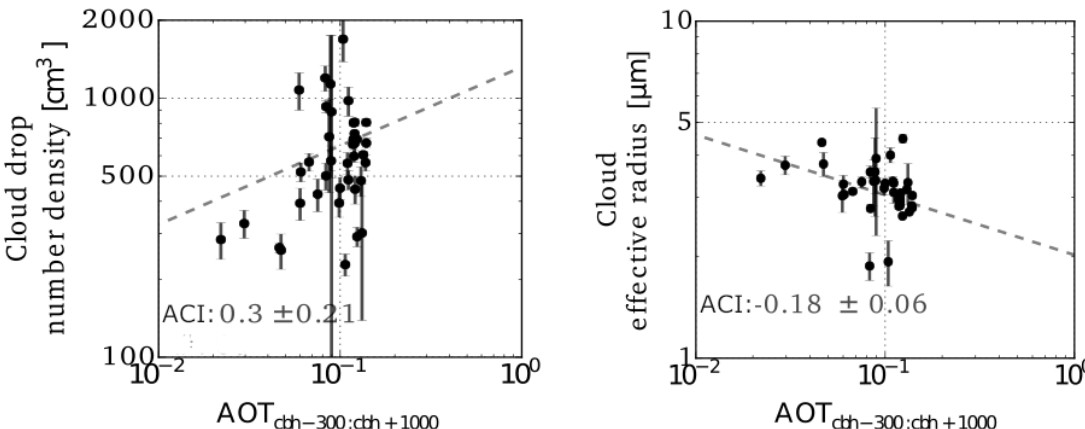

**Figure 7.** (Left) Weighted mean of the cloud drop number density versus daily average AOT for each cloud selection. (Right) Weighted mean of the cloud effective radius versus daily average AOT for each cloud selection. For both cloud properties a linear fit is plotted and the ACI is given. The standard deviation was used as weights in the fit.

cloud effective radius is at the high end of values found by previous studies. For example, McComiskey et al. (2009) found values of $|ACI|_r$ ranging from $0.0 - 0.16$ in marine stratus clouds, while Kim et al. (2008) found values from $0.04 - 0.17$ in continental stratus. Higher values ($0.13 - 0.19$) were found in the Arctic (Garrett et al., 2004), and for very large ranges of aerosol concentration including strong pollution ($0.21 - 0.33$) (Ramanathan et al., 2001).

### 3.3 Aerosols below the cloud

In order to get aerosol and cloud proxies closer together in time, $ACI_r$ and $ACI_N$ were also calculated using the aerosol extinction below the clouds during cloudy periods. For this, the aerosol extinction profile was computed using Eq. (A8), but with the normalisation height set inside the cloud and the extinction-to-backscatter ratio set to 20 sr in the cloud and 50 sr below the cloud, as described in A3.2. Furthermore, the cloud extinction-to-backscatter ratio was corrected for multiple scattering using Eq. (A10). The extinction profile was determined from 200 m above the lidar, to avoid overlap, until 300 m below cloud base to avoid the mixing region of wet aerosols just below the cloud. The mean aerosol extinction coefficient was used instead of the AOT, because the height of the range bins changed per cloud selection. Cloud retrievals of 30 s intervals were averaged, with a minimum of 3 values and a maximum of 24 values, corresponding to cloud periods of 1.5 to 12 minutes. The errors from the lidar inversion were used as weights in the determination of the ACI-values. The results for the 2016 measurements are plotted in Fig. 8.

The ACI for all cloud periods during the 2016 campaign show varying results. Many values are beyond the theoretically feasible values, indicated in the plots by the grey horizontal lines. Theoretically, the absolute value $|ACI|_N$ must be below 1 and the absolute value $|ACI|_r$ below 0.33 (McComiskey et al., 2009), reaching the maximum absolute values if all aerosol particles are activated to droplets. However, a number of retrievals show much larger values, characterised by large uncertainties. The

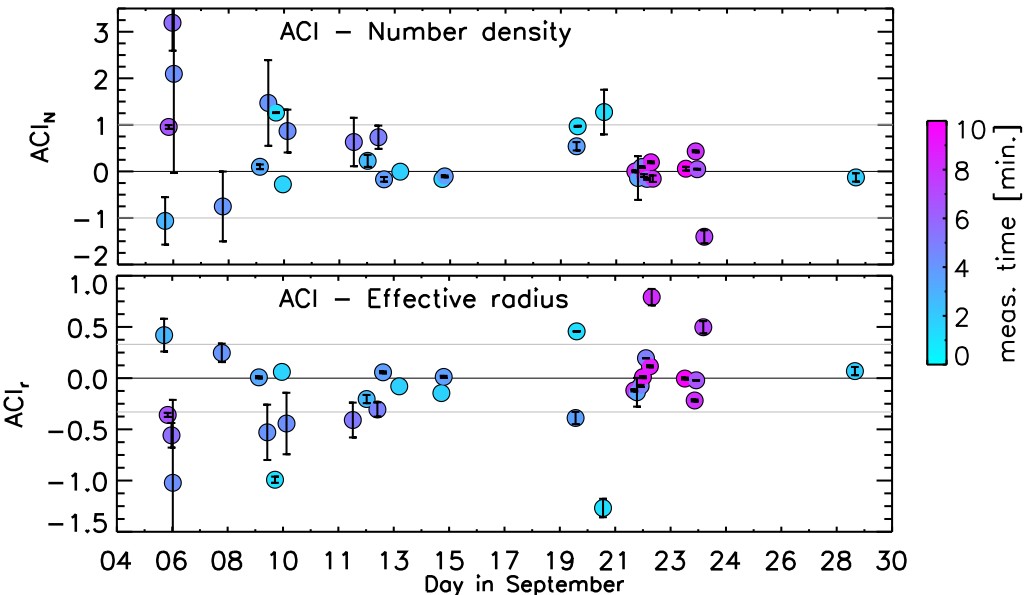

**Figure 8.** Aerosol-cloud interaction (ACI) for each selected cloud period in Sept. 2016, using the average aerosol extinction profile below a cloud and the retrieved cloud droplet number density (top) and the retrieved cloud droplet effective radius (bottom). The error bars indicate the standard deviation of the measurements during each selected interval, the colors indicate the duration of the intervals. The grey horizontal lines indicate the physically feasible bounds of the ACI values.

theoretical numbers are based on idealized clouds in a constant atmospheric state. The retrievals with large numbers and large uncertainties must be associated with variable meteorological conditions that drive the changes in cloud and aerosol properties.

Around 12–15 Sept. and 21–24 Sept. $ACI_N$ and $ACI_r$ are mostly within the physical ranges with small uncertainties. These episodes correspond to periods of increased AOT over Ascension Island, see Fig. 3. This suggest that during those periods the interaction of smoke with the cloud base is the driving mechanism for forming more numerous, smaller droplets.

## 4   Discussion

The three presented methods all suggest some indication of the Twomey effect in the cumulus clouds around Ascension in 2016 during various episodes. However, changing meteorological conditions could affect the results. An inspection of (back)trajectories during the measurement period showed that the MBL around Ascension Island is very persistent, c.f. Fig 4. Daily backtrajectories of air ending at 600 m altitude over Ascension Island invariably showed MBL air being transported from the southeast with little to no vertical displacement for all the days during the 2016 campaign, indicating a stable of moist

well-mixed air in the MBL as expected over this region. On the other hand, the air transported to the cloud layer, e.g. at 2100 m altitude, was from the east most of the time (loaded with smoke), but also from the west and variable.

In a recent paper, Ryoo et al. (2022) shows that during Sept. 2016 the Southern African Easterly Jet increases develops from weak in the beginning of Sept. 2016 to strong at the end of the month, with increased relative humidity and BC concentrations over central south Atlantic Ocean at 600 hPa especially around 15–17 and 27 Sept. 2016. These episodes correspond to the increased AOT in Fig. 3, showing the dominance of the large scale circulation in the free troposphere on the AOT fluctuation over Ascension Island. However, the correlation between BC concentration and relative humidity can also explain a positive correlation between the AOT and cloud droplet number density, as observed in Figs. 6 and 7, if more particles become activated with more available moisture. However, in that case the observed reduction of cloud effective radius is unlikely, and we conclude that the advection of smoke from the African continent reduces the effective cloud droplet size at cloud base through the first aerosol indirect effect.

## 4.1 Cloud parameters

Lidar retrievals of the cloud parameters have been performed in only a few cases before (Donovan et al., 2015; Sarna and Russchenberg, 2016; Jimenez et al., 2020). Below, the cloud retrievals from the UV-polarisation lidar are compared to retrievals from cloud radars located on the ARM Research Facility. Unfortunately, in 2016 the cloud radar was operational only for a short period during the campaign, so 2017 data are also used to assess the cloud data from the lidar retrievals.

In 2016, a W-band Scanning ARM Cloud Radar (WSACR) was operated from the start of the lidar measurement period until 11 September. In 2017, a KA-band Scanning ARM Cloud Radar (KASACR) was operated during the entire period of the lidar operation. WSACR was operated at a frequency of 94 GHz and KASACR at 35.3 GHz. Both radars have a field of view of 0.3 degrees. Although the radars were operated with scanning strategies, here only the vertical pointing modes were used, taken each hour for a duration of 4 minutes. The 2D radar reflectivity factor $Z$, with a time resolution of 2 s and a vertical resolution of 30 m, was collected from the ARM website.

The radar reflectivity was used to derive $R_{\text{eff}}^{100}$ following the method described by Frisch et al. (1995): Assuming a cloud with a lognormal droplet size distribution

$$n(r) = \frac{N_d}{\sqrt{2\pi}r\sigma_x} \exp\left(\frac{-(\ln(r) - \ln(R_0))^2}{2\sigma_x^2}\right),$$ (3)

where is $R_0$ the median radius and $\sigma_x$ the spread of the lognormal distribution, the effective cloud droplet radius $R_{\text{eff}}$ is related to the median radius by

$$R_{\text{eff}} = R_0 \exp(\frac{5}{2}\sigma_x^2),$$ (4)

and the radar reflectivity is

$$Z = 2^6 N_d R_0^6 \exp(18\sigma_x^2).$$ (5)

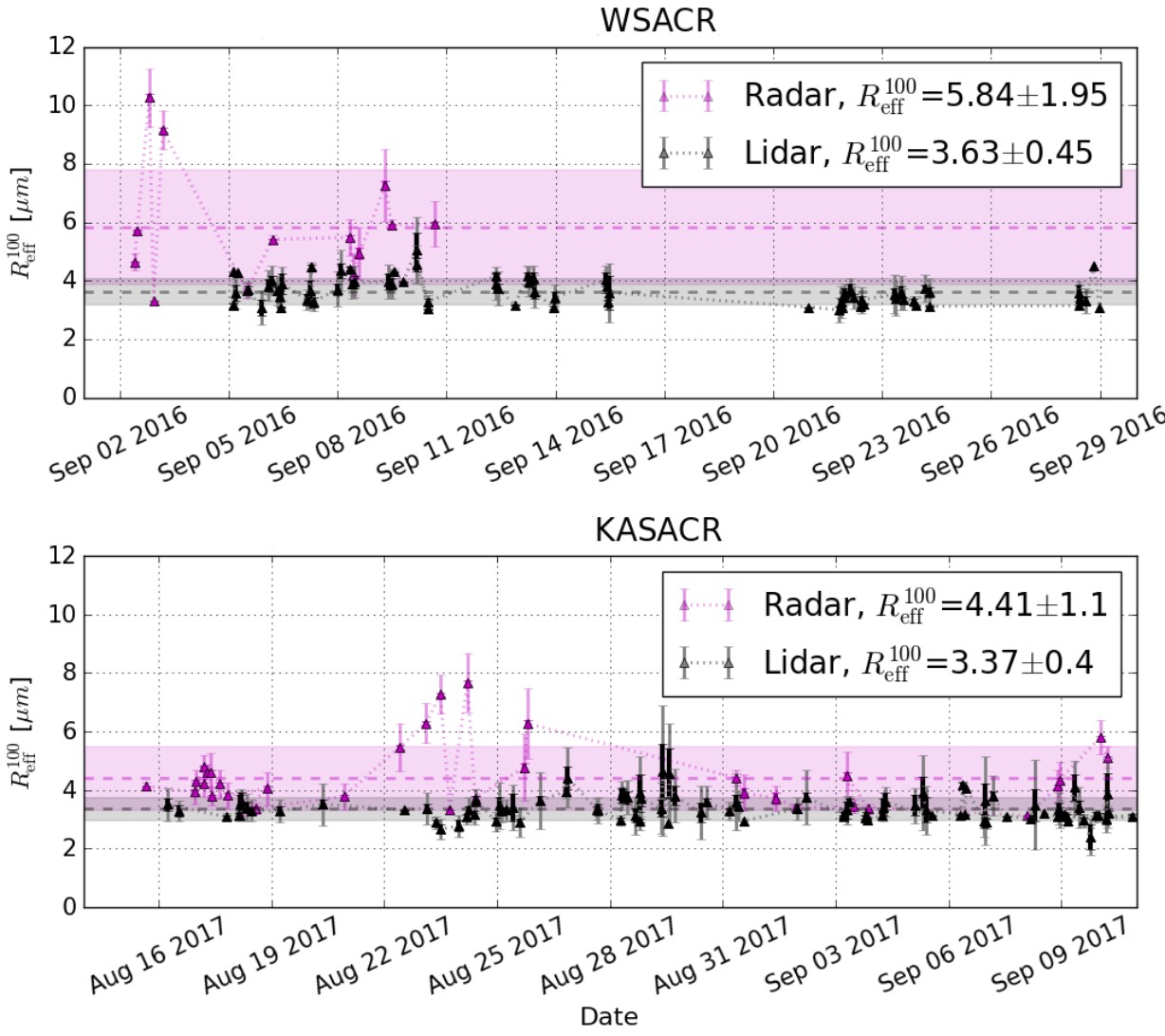

**Figure 9.** $R_{\text{eff}}^{100}$ for selected cloud periods in 2016 (top panel) and 2017 (bottom panel) from the lidar (grey) and the cloud radar (purple). The shading shows the standard deviation or retrieval error, while the variance in the cloud per measurement period is given by the error bars. The dashed line gives the mean $R_{\text{eff}}^{100}$.

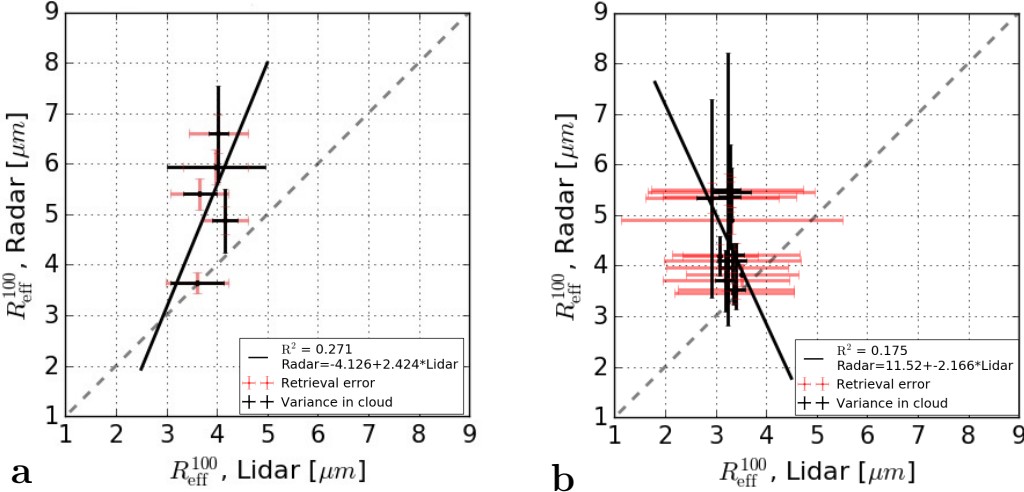

**Figure 10.** Comparison of daily averaged $R_{\text{eff}}^{100}$ from lidar and cloud radar in (a) 2016 and (b) 2017. The retrieval error is shown by the black error bars, while the variance of the daily measurements is shown by the red error bars. The dashed line shows the 1:1 line, and the black full line a linear least squares fit. The slope and offset of the fit are indicated in the legend, along with Pearson's correlation coefficient.

This gives a relationship for the effective cloud droplet radius

$$R_{\text{eff}}(z) = \frac{1}{2} \left( \frac{Z(z)}{N_d} \right)^{1/6} \exp(-0.5\sigma_x^2). \tag{6}$$

and the value of $R_{\text{eff}}^{100}$, to be compared to the lidar retrievals is simply given by the above equation with z corresponding to 100
meters above cloud base with the cloud base supplied by co-located lidar ceilometer measurements (see App. B). The value for
$\sigma_x$ was set to $0.34 \pm 0.09$, which is a typical value for marine, low-level clouds (Fairall et al., 1990; Frisch et al., 1995; Miles
et al., 2000). An uncertainty of $\pm$ 3 dBZ in the reflectivity factor was used to compute the error margins. $N_d$ can be estimated
from the lidar inversions, see Eq. (A11). Daily averaged lidar estimates of $N_d$ were around $466 \pm 127$ cm$^{-3}$ in 2016 and $540 \pm$
$142$ cm$^{-3}$ in 2017. The uncertainty of retrieved $Nd$ is between 25% and 50% (Donovan et al., 2015). The lidar estimates of $N_d$
is higher than earlier reported values of $100 \pm 70$ cm$^3$ for marine, low-level clouds (Davidson et al., 1984; Martin et al., 1994)
and used by Frisch et al. (2002). However, $N_d$ is seasonally dependent, with higher values in the boreal summer over SEAO
(Li et al., 2018) and western North Atlantic Ocean (Dadashazar et al., 2021), and Eq. 6 shows that relatively large changes in
$N_d$ will produce only small changes in $R_{\text{eff}}$. The use of the literature value of $100$ cm$^3$ in the radar estimates increased the
effective radius by about 3 microns.
$R_{\text{eff}}^{100}$ estimates from lidar and cloud radar are compared in Figs. 9 and 10. Figure 9 shows the lidar retrievals for selected
cloud periods, with the variance in the measurements shown by error bars and the estimated measurement error shown by
the shaded purple (radar) and grey (lidar) areas. The average retrieved effective droplet radii (shown by the dashed lines) was
$3.63 \pm 0.45\mu$m in 2016 and $3.37 \pm 0.4\mu$m in 2017 for the lidar retrieval, and $5.84 \pm 1.95\mu$m in 2016 and $4.41 \pm 1.1\mu$m in 2017

for the radar retrievals. Figure 10 shows scatter plots of the daily averaged retrievals of $R_{\text{eff}}^{100}$ from lidar and radar retrievals in 2016 and 2017. In general, the estimates of $R_{\text{eff}}^{100}$ from the cloud radar are larger than from the lidar. This will be even larger for lower values for $N_d$. The comparison is complicated by the low number of measurements in 2016. In 2017, the average value is closer, but the alignment problems complicates the comparison and correlation was found between the radar and lidar estimates.

The dependence on the assumed value of $N_d$ can be removed altogether using cloud liquid water path (LWP) data from a microwave radiometer (MWR) (Frisch et al., 2002). An MWR was operated at 23.8 and 31.4 GHz alongside the WSACR until 11 September 2016. The method radar+MWR method described in (Frisch et al., 2002) was also applied in addition to the radar-only (Frisch et al., 1995) method. The radar+MWR method however, tended to yield particle size measurements much higher than the radar only approach. Moreover, the radar+MWR results tended to yield $R_{\text{eff}}^{100}$ values strongly inconsistent with non-drizzling clouds (e.g. values greater than 15 $\mu$m) and unrealistically low values of number density (e.g. less than 5 cm$^{-3}$). The reason(s) for this are unclear but may point to biases in the LWP data used or an error in the implementation.

The differences in effective radius retrieval could be the consequence of a number of factors. Both the radar-only and radar+MWR methods are sensitive to the presence of drizzle, while the lidar-only method is relatively insensitive to the presence of drizzle (Donovan et al., 2015). Even small amounts of drizzle may result in radar reflectivity based retrievals overestimating cloud particle sizes (e.g. Fox and Illingworth, 1997; Küchler et al., 2018; Wang and Geerts, 2003). It should be noted, however, that the smaller of effective radius seen with lidar is consistent with that reported by Jimenez et al. (2020) (e.g. Figure 6). Also, Conant et al. (2004) report cloud droplet effective radii from cloud radar measurements in warm cumulus clouds growing from about 2 $\mu$m near cloud base to 10 $\mu$m at 1000 m above cloud base. Frisch et al. (2002) report radar estimates of $R_{\text{eff}}^{100}$ in stratus clouds ranging from 4 $\mu$m to 8$\mu$m in close agreement with aircraft measurements, depending strongly on cloud height.

## 5   Conclusions

In this study, aerosol-cloud interactions were studied in the broken cloud deck over Ascension Island during the African monsoonal dry season in 2016 and 2017, which is about July to October. During these months, plumes of smoke from vegetation fires drift over the ocean. The typical clouds over Ascension Island are cumulus clouds in the terminating stage of the stratocumulus to cumulus (SCT) transition. Smoke affects this transition is various ways. We found that the presence of smoke decreases the cloud droplet sizes near the cloud base and increases the cloud droplet number density, likely due to the first aerosol indirect effect. On average, the cloud drop number density was 294$\pm$91 cm$^3$ and cloud effective radius 3.81$\pm$0.6 $\mu$m during smoke free days, compared to 611$\pm$191 cm$^3$ and 2.85$\pm$0.2 $\mu$m during days with smoke at cloud level. Similarly, aerosol-cloud interactions were quantified using cloud base parameters during cloud periods and daily averaged AOT at cloud level: the cloud droplet number density ACI$_N$ was $0.3 \pm 0.21$ cm$^{-3}$ and the cloud effective radius ACI$_r$ was -0.18 $\pm$0.06 $\mu$m.

Lastly, aerosol and cloud properties were retrieved simultaneously by the lidar during cloudy periods. This was possible by retrieving aerosol extinction profiles under the clouds. During two episodes, 12–15 Sept. 2016 and 20–24 Sept. 2016 an

indirect effect was found, corresponding to periods with increased transport of air from the African continent over the SEAO. This increased both the BC concentration and AOT over Ascension, but also the relative humidity. However, the results show a decrease of droplet size and increase of droplet number density near the cloud base related to increases in aerosol concentration, suggesting that the smoke is responsible for more numerous but smaller cloud droplets, which will shorten the SCT, both by warming the MBL during the day and making cloud droplets more susceptible to evaporation.

The lidar retrieved values of the effective radius were small compared to many other studies of cloud droplet sizes of warm low level clouds. However, lidar estimates of cloud droplet effective radius are restricted to cloud base values, and care should be taken when comparing estimates from ground-based radars and satellite retrievals. Vertical profiles of $R_{eff}$ are typically strongly growing from a few microns to over ten microns and more until the cloud top. Radar beam can penetrate the cloud completely and the average retrieved effective radius depends on the assumed vertical distribution. Satellite retrievals of cloud droplet sizes are typically biased to the cloud-top retrievals. Therefore, comparisons between these types of retrievals should be performed only when corrected for the vertical profile of the cloud droplet sizes (Zhang et al., 2011). A comparison with radar estimates of droplet sizes near the cloud base showed consistent values, to within the measurement uncertainties.

This is the first time a UV-polarisation lidar was used to determine cloud parameters at the cloud base of marine cumulus clouds in the SCT zone over the SEAO. The measured depolarisation of the lidar beam was fitted to LUTs of precalculated depolarisation by cloud droplets using Monte Carlo (MC) simulations, relating the depolarisation to cloud droplet effective radius and the cloud extinction parameter at a reference height using a proper cloud model. This method shows potential for the monitoring of aerosol-cloud interactions at strategically positioned locations in climate sensitive areas, like the SEAO. The simultaneous retrievals of aerosol extinction and cloud properties from one single instrument can be helpful in the measurement of aerosol indirect effects, which constitutes the largest uncertainties in global climate models. However, we found that proper calibration of the instrument and careful selection of the data are essential.

*Data availability.* The UV-polarisation lidar data acquired on Ascension Island are available on the KNMI open data platform. DOI: https://doi.org/10.21944/5qqy-0c37

**Appendix A: Theory**

The theory of the applied methods has been described in earlier papers, cited in the text. For completeness, the method applied to the UV-lidar data on Ascension Island is described below.

**A1   UV-lidar**

The total power returned to a lidar by backscattering in the atmosphere under single-scattering conditions is

$$P(z) = \frac{C_{\mathrm{lid}}}{z^2} \beta_\pi(z) \exp\left(-2 \int_0^z \alpha(z')dz'\right) \tag{A1}$$

where $P$ is the power received by the instrument, $z$ the altitude from the instrument along the line of sight, $C_{\mathrm{lid}}$ the lidar calibration coefficient, $\alpha$ the atmospheric extinction coefficient, and $\beta_\pi$ the atmospheric backscatter coefficient. The atmospheric extinction and backscatter coefficients can be divided into a molecular, aerosol, and cloud part, viz.

$$\alpha = \alpha_{\mathrm{m}} + \alpha_{\mathrm{a}} + \alpha_{\mathrm{c}}$$
$$\beta_\pi = \beta_{\mathrm{m}} + \beta_{\mathrm{a}} + \beta_{\mathrm{c}} \tag{A2}$$

The extinction-to-backscatter ratio, or lidar ratio, $S$ is defined as $S(z) = \alpha/\beta$. The aerosol scattering ratio ($R_{\mathrm{asca}}$) is defined as $R_{\mathrm{asca}} = (\beta_{\mathrm{a}} + \beta_{\mathrm{m}})/\beta_{\mathrm{m}}$, which is 1 if there are no aerosols.

## A2   Molecular scattering

The molecular backscatter coefficient can be calculated using (Collis and Russel, 1976)

$$\beta_{\mathrm{m}} = \frac{\rho_{\mathrm{air}}}{M} \left( \frac{\lambda}{550} \right)^{-4.09} 10^{-32}\ \mathrm{m}^{-1}\mathrm{sr}^{-1}, \tag{A3}$$

where $\lambda$ is the wavelength, $M$ is the average molecular mass of air ($4.81 \cdot 10^{-26}$ kg), and the atmospheric density was determined using

$$\rho_{\mathrm{air}} = \frac{p}{T} \frac{1}{\mathrm{R_{dry\,air}}}, \tag{A4}$$

where $p$ is the measured pressure, $T$ the measured temperature and $\mathrm{R_{dry\,air}}$ the gas constant for dry air with an average value of 287 $\mathrm{J\,kg}^{-1}\mathrm{K}^{-1}$. The temperature and pressure were determined from radiosondes, launched four times daily from Ascension
Island. The molecular extinction coefficient $\alpha_m$ can be calculated using the molecular extinction-to-backscatter ratio $S_{\mathrm{mol}} = 8\pi/3$ sr (Guzzi, 2008). At the lidar wavelength of 355 nm molecular scattering is strong and this was used to calibrate the lidar. Details can be found in Schenkels (2018).

## A3   AOT retrieval

For a lidar operating in the UV, molecular scattering is strong and must be accounted for in the inversion. In this case, a two-
mode method following e.g. Klett (1981) and Fernald (1984) can be applied using transformed variables (Sarna et al., 2021)

$$P'(z) = S(z)P(z)\exp\left( 2\int\limits_{0}^{z} \alpha_m(z') - S(z')\beta_m(z')dz' \right) \tag{A5}$$

and

$$\alpha'(z) = \left( S(z)\beta_m(z) + \alpha_a(z) \right). \tag{A6}$$

Now Eq. (A1) can be rewritten as

$$P'(z) = \frac{C_{lid}}{z^2}\alpha'(z)\exp\left(-2\int_0^z \alpha'(z')dz'\right),$$   (A7)

with the analytical solution

$$\alpha'(z) = \left[\frac{\frac{P'(z)z^2}{P'(z_0)z_0^2}}{\frac{1}{\alpha'(z_0)} + 2\int_z^{z_0} \frac{P'(z)z^2}{P'(z_0)z_0^2}dz'}\right].$$   (A8)

where $z_0$ is a normalisation height. From the transformed variable $\alpha'$, the aerosol extinction is derived to be $\alpha_a(z) = \alpha'(z) -$
$S(z)\beta_m(z)$. The aerosol backscatter coefficient is now derived by dividing the aerosol extinction by the height dependent lidar ratio. The aerosol optical thickness ($\tau$) of a layer can be obtained by integrating the aerosol extinction profile over the altitude of the layer:

$$\tau(z_1; z_2) = \int_{z_1}^{z_2} \alpha_a(z)dz$$   (A9)

### A3.1   Cloud-free scenes

In clear sky scenes the normalisation height is set to an altitude at which the aerosol extinction is zero. From literature (e.g. Wandinger et al., 2016; Greatwood et al., 2017) and from observations on the island, it was concluded that marine aerosols are always present in the lower boundary layer, up until 1200 m. $S_{\mathrm{marine}}$ was set to be 25 sr, a good approximation for marine aerosols (Wandinger et al., 2016; Cattrall et al., 2005; Müller et al., 2007). (Aged) smoke and dust were often, almost always, present above the boundary layer, in the layer from 1200 m to 5000 m, sometimes mixed in the boundary layer. For this layer
the lidar ratio $S_{\mathrm{dark}}$ was set to 50 sr (Wandinger et al., 2016). Above 5000 m, the air was mostly clean and clear of aerosols and the lidar ratio reduces to the molecular extinction-to-backscatter ratio defined above. The normalisation height was set to 7 km. Various tests were performed varying $S_{\mathrm{marine}}$ and $S_{\mathrm{dark}}$ around their values of 25 and 50 sr to check the sensitivity of the choices, resulting in 5% changes in AOT within the expected reasonable ranges of S.

### A3.2   Aerosol below clouds

In order to derive aerosol optical thickness close to clouds, aerosol extinction profiles were retrieved for cloudy scenes under the clouds, using Eq. (A8). However, in this case the normalisation height is not located at an altitude without aerosols, but inside the cloud where the aerosol contribution can be neglected. The normalisation height was determined by the cloud base height and the cloud extinction. The extinction-to-backscatter ratio was set to 20 sr in the cloud and 50 sr below the cloud (Wandinger et al., 2016).
Furthermore, multiple scattering, which influences the lidar return and the cloud extinction, should be taken into account in a cloud. Therefore, the cloud extinction-to-backscatter ratio, used to determine $\alpha'$ in Eq. (A8), was corrected by a multiple scattering correction factor $\eta$

$$S_c = \frac{(1 - \eta)\alpha_c}{\beta_c}. \tag{A10}$$

The correction factor $\eta$ was determined from a sensitivity study over three days in 2016 with broken clouds. Aerosol profiles
below clouds during these days were fitted to aerosol retrievals during clear sky spells close in time on these days. The
correction factor was varied between 0.3 and 0.5 in steps of 0.05, resulting in overcorrection and undercorrection. The best fit
was found for 0.35 and 0.4. The difference in aerosol extinction coefficient at an altitude of 300 m below cloud base between $\eta$
= 0.35 and 0.4 is about $2.6 \cdot 10^{-5}$ m$^{-1}$. In all subsequent processing a value of $\eta = 0.4$ was used. See Tenner (2017) for details.

## A4   Clouds

Although the lidar equation (A1) formally only applies for single scattering, the derivation of cloud extinction and backscatter
coefficient in this section is based on a polarisation change after multiple scattering, first developed by Donovan et al. (2015).
Light returning from a liquid cloud will be partially depolarised due to multiple scattering by the cloud droplets (Liou and
Schotland, 1971). This multiple scattering in a liquid water cloud can be simulated by a Monte Carlo (MC) model, assuming
a cloud model. This was achieved using the Earth Clouds and Aerosol Radiation Explorer (EarthCARE) simulator (ECSIM)
lidar-specific MC forward model. The ECSIM lidar MC model is a modular multi-sensor simulation framework, which was
used to calculate the spectral-polarisation state of the lidar signal.

The underlying cloud model is based on clouds with a linear liquid water content (LWC) profile from cloud-base and a
constant cloud droplet number density ($N_d$) (e.g. de Roode and Los, 2008). Various MC simulations were carries our for
different LWC slopes, number densities and lidar field-of-views, and cloud base values. The MC results were then used to
product look-up-tables which form the basis of a forward model which is fast enough to serve as the forward retrieval model
in an optimal-estimation retrieval procedure. Details are described in the remainder of this section.

The cloud droplet size distribution was defined as a single-mode modified-gamma distribution (Miles et al., 2000)

$$n(r) = \frac{N_d}{R_m} \frac{1}{(\gamma - 1)!} \left(\frac{r}{R_m}\right)^{\gamma - 1} \exp\left(\frac{r}{R_m}\right), \tag{A11}$$

where $N_d$ is the cloud droplet density, defined to be constant with height, $r$ is the droplet radius, $R_m$ the mode radius and $\gamma$ the
shape parameter of the distribution.

A linear liquid water content defines a constant liquid water lapse rate, $\Gamma_l$. When the liquid water content increases with
height and the number density remains constant, the cloud droplet effective radius, defined as

$$R_{\text{eff}} = \frac{\int n(r) r^3 dr}{\int n(r) r^2 dr}, \tag{A12}$$

will increase with height. The cloud extinction coefficient, $\alpha_c$, also increases with height. This leads to the prediction that
the depolarisation ratio is generally increasing throughout the cloud, while observations show that the depolarisation ratio

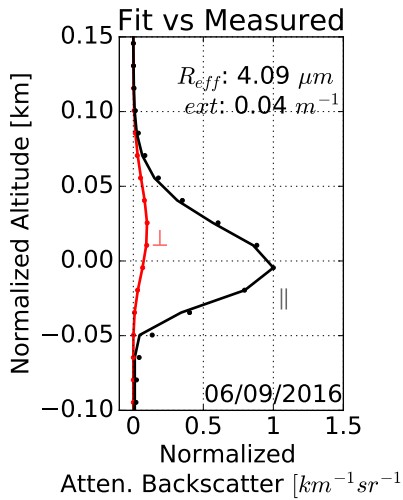

**Figure A1.** Measured (solid line) and fitted (dots) vertical profiles for the parallel attenuated backscatter (black), perpendicular attenuated backscatter (red) on 6 Sept. 2016, for the selected cloud (C) in Fig. 2.

may exhibit a peak (Sassen and Petrilla, 1986). Furthermore, the model represents semi-infinite clouds, with a cloud top at infinity. However, the lidar signal can only penetrate a few hundred meters into the cloud. Therefore, no information is known about the upper part of the cloud and any retrieved parameters are only applicable to the cloud-base region and the parameters were calculated for a reference height. In this research, 100 m above cloud-base was assumed. This simple but effective cloud

representation reduces the parameters to describe the cloud to two, the cloud extinction $\alpha_c^{100}$ at reference height, and the cloud effective radius $R_{\mathrm{eff}}^{100}$ at reference height.

MC model simulations were performed for various values of the cloud base height (CBH), the lidar field-of-view (FOV), $R_{\mathrm{eff}}^{100}$ and the adiabatic cloud-base liquid water lapse-rate $\Gamma_l$. The values are replicated from (Donovan et al., 2015) in Table A1. Look-up tables (LUTs) were generated from the simulations and predefined input parameters, the lidar constants and initial

values for $R_{\mathrm{eff}}^{100}$ and $\alpha_c^{100}$. These LUTs contain information on the simulated parallel and perpendicular attenuated backscatter and therefore the depolarisation ratio.

**Table A1.** Range of parameters used in the ECSIM MC calculations

| Parameter | Values |
|---|---|
| CHB[km] | 0.5,1.0,2.0,4.0 |
| FOV[mrad] | 0.5,1.0,2.0,4.0 |
| $R_{\mathrm{eff}}^{100}$ [$\mu$m] | 2.0,2.6,3.3,4.3,5.6,7.2,9.3,12.0 |
| $\Gamma_l$ [g m$^{-3}$ km$^{-1}$] | 0.1,0.2,0.4,0.6,0.8,1.0,1.2,1.4,1.6,1.8,2.0 |

The observed attenuated backscatter and depolarisation ratio were compared to the LUTs to find the best matching values for $R_{\text{eff}}^{100}$ and $\alpha_c^{100}$, by iteratively minimizing a cost-function (Rodgers, 2000). In Fig. A1, the observed and fitted attenuated backscatter profiles from the LUTs are shown, for a cloud selected on 26 August 2017. The dotted lines correspond to the fitted values from the LUTs, with the parallel attenuated backscatter in black, the perpendicular attenuated backscatter in red and the depolarisation ratio in magenta. The observed profiles are represented by the corresponding solid lines.

The cloud drop number density $N_d$ follows from the cloud effective radius and the cloud extinction

$$N_d = \alpha_c^{100} \frac{1}{2\pi} \frac{1}{(R_{\text{eff}}^{100})^2} \frac{1}{k}, \tag{A13}$$

where $k$ is $0.75 \pm 0.15$.

Because multiple-scattering occurs in a cloud, the LUTs, the shape of the attenuated backscatter and the depolarisation ratio profiles are all well-defined functions of the LWC and effective radius profile. For single-scattering the parallel attenuated backscatter profile will not depend on the effective radius profile.

It is important to note that the CBH is difficult to define from real observation due to the presence of sub-cloud drizzle and the presence of growing aerosol particles. The MC based inversion results would be very sensitive to the absolute calibration of the attenuated backscatter if the CBH is used as a reference. Therefore, the peak of the observed parallel lidar attenuated backscatter is used as a reference instead of the CBH in the fitting procedure. Consequently, the CBH is produced as a by-product and in Sect. 3 the derived CBH will be compared to observations of the CBH using different instruments.

## Appendix B: Cloud Base Height validation

It is important to compare the cloud parameters from the lidar and the cloud radars at the same relative height, since the effective radius strongly depends on the height in the cloud. The effective radius was determined at a reference of 100 m above cloud base height (CBH), which was related to the peak of the observed parallel lidar attenuated backscatter. The accuracy of the backscatter peak as the CBH cannot directly be compared to the CBH from the cloud radar, because of the different locations of the instrument. The effect of the spatial distance between the instruments was investigated by comparing CBH from two ceilometers that were installed in the airport and the ARM main site. This is illustrated in the left panel Fig. B1 for one day, 26 August 2017.The CBH from these instruments, relative to the mean sea level, are highly correlated in general (Pearson's correlation coefficient was 0.931). However, on average a higher cloud fraction was found over the main ARM site compared to the airport, due to the higher elevation of the site. More low-level clouds were detected over the ARM main site and the cloud fraction differed. However, this should not affect the analyses too much, since the main difference is in the low-level clouds and the selected cloud periods had CBH's higher than 1000 m.

The CBH from the lidar and from the ceilometer at the airport were compared, shown in the right panel of Fig. B1. The correlation was higher than 90%. Therefore, the relative height of the peak of the backscatter can be considered a good proxy for the relative position of the CBH.

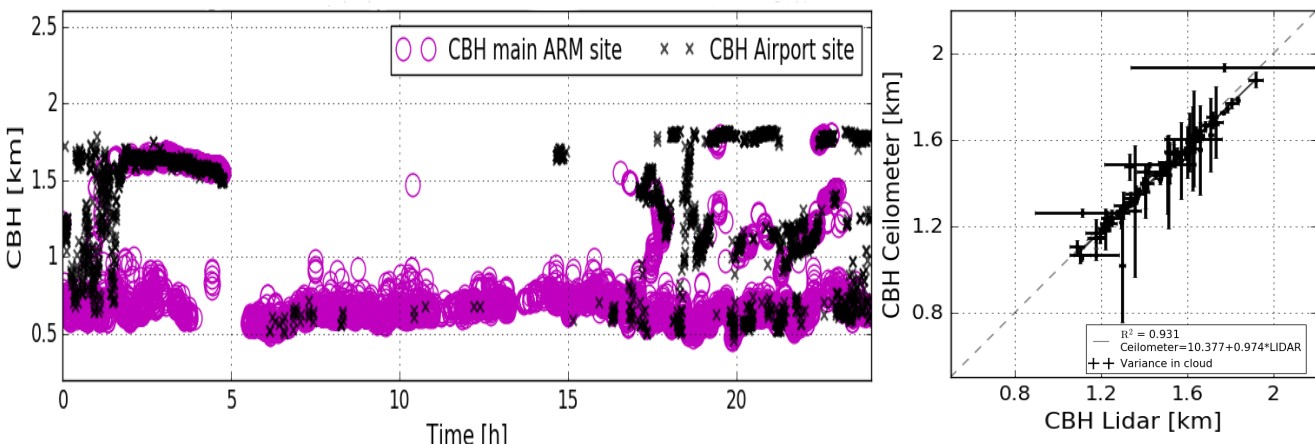

**Figure B1.** (left) The CBH from the ceilometer at the airport (black crosses, elevation 79 m) compared to the CBH from the Ceilometer at the main ARM site (purple circles, elevation 365 m) on 26 August 2017. The CBH is measured relative to the mean sea level. (Right) Comparison of the cloud base height determined from the UV-lidar and the ceilometer located on the airport. The dashed line is the 1:1 line.

*Author contributions.* MdG authored the science application, coordinated and managed the measurement campaigns, and wrote the paper; KS co-authored the science application; JB performed the 2016 measurements; ET analysed the 2016 data; MS performed and analysed the 450 2017 measurements; DPD designed the MC model, calibrated the UV-lidar and overlooked the science.

*Competing interests.* None

*Acknowledgements.* This project was financed by the Pieter Langerhuizen Stipendium of the Koninklijke Hollandsche Maatschappij der Wetenschappen (http://www.khmw.nl/ in Haarlem, The Netherlands, supplemented with financial support from TUD and manpower from TUD, KNMI and WUR, for which we are greatly indebted. We are also grateful for the support of the RAF personnel and staff at Wideawake 455 airfield. Special thanks go to Prof. Jim Haywood from the UK Met Office and Exeter U. for his leadership and help on numerous occasions of logistical mayhem, and Jenna Macgregor of the Ascension Island Met Office for her initiatives and help on the Ascension side.

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
