# Peer review of "Aerosol first indirect effect of African smoke at cloud base of marine cumulus clouds over Ascension Island, south Atlantic Ocean"

_Atmospheric Chemistry and Physics, 2022_

## Referee Comment (RC1)

**Review of "Aerosol first indirect effect of African smoke in marine stratocumulus clouds over Ascension Island, south Atlantic Ocean" by M. de Graaf et al.**

This study uses a single instrument to study aerosol first indirect effect with one month of data collected at the Ascension Island in the middle of the south Atlantic Ocean during the southern African biomass burning season. The manuscript is relatively well written, and the scope of the study is properly fitted for ACP, however, the current form of the manuscript reads too technical (more suited for AMT), and the significance of the results shown needs to be assessed before it can be considered for publication. Moreover, upon addressing the above mentioned points, the structure of the manuscript also needs to be revised for the ease of readership.

**Main concerns:**

1. Statistical significances of the analyses (all 3 methods associated with Fig. 4-6) need to be included and discussed. Working with only one month of available can be quite challenging, but I believe one can still make valuable statements with proper significance assessments.
2. Attribution of the Twomey effect requires constant cloud macrophysical properties, e.g., cloud LWP, and environmental conditions, I understand this can be difficult with less than 40 sample size, but these limitations/assumptions need to be acknowledged when interpreting the results and making attributions (i.e., saying these indicate the Twomey effect).
3. Section 5.2 confuses me, if one cannot validate the representativeness of the retrieved cloud properties, how can one study the interactions of them with retrieved aerosol properties? Moreover, if the Radar retrievals of cloud properties is really problematic and biased ($r_e$ retrieved using daily averaged or some assumed Nd values are certainly not suited for aerosol-cloud interactions studies), what's the point of validating Lidar Nd with Radar retrievals? Furthermore, it doesn't validate a retrieved variable when another variable retrieved using the same instrument is involved in the validation.
4. The current form of the manuscript reads too technical, especially Sections 2.1-2.4 and 5.2. These technical details can be condensed and summarized in the main text, with details provided in an appendix or a supplement.
5. The validation section needs to be moved up before showing the results. How can a reader interpret these results without knowing the retrievals that these statements based on are validated?
6. The current form of the Conclusions reads like a summary and repetitive of what have been stated. Emphasizing on the advantages (and caveats) and implications of the study would be very helpful.

**Minor comments:**

The authors tend to state existing knowledges without providing references, for example:

1) P1 Line17 and P8 Line166, the typical thermodynamical structure of the MBL clouds over Ascension during the dry season can be found in Zhang & Zuidema 2019 ACP.
2) P2 Line33 and P8 Line173, Zuidema et al. 2018 GRL provides a more updated overview on LASIC and some first results.

3)  When providing information on the transport, seasonality, and distribution of the smoke aerosols to set up the context on the complex environment within which smoke-cloud interactions manifest over the SE Atlantic during the southern African biomass burning season, Adebiyi & Zuidema 2016 QJRMS, Adebiyi et al. 2015 JClimate, and Zhang & Zuidema 2021 are suitable references.
4)  P11 L222, reference for the theoretically feasible values?
5)  P13 L284, "from the literature", which one?

Since cloud properties retrieved by the lidar only represents cloud base values, and (as the authors also mentioned) cloud droplet size is highly dependent on height, I wonder if it's more appropriate to indicate that this study focuses on the aerosol indirect effect at cloud base in the title, with"… at cloud base of marine stratocumulus clouds …"?

The current introduction is too thin, introduction to existing knowledges on aerosol cloud interactions over the region is needed to set up the scientific question. The introduction to the Ascension Island and its environment, i.e., the smoky SE Atlantic during dry season, and the campaign info need to be moved up, preferrable to the introduction. When reading the current manuscript, a reader has no idea of the context (the condition under which these measurements were made) until P7 Section 3.

If the 2017 measurements were affected by alignment problems, why all your proof of concept exemplary figures show 27 Aug 2017? Why not use a day from 2016?

Why the sample sizes of the 3 methods not consistent? (37 in Fig. 4, 39 in Fig. 5, and 32 in Fig. 6). This needs to be justified.

Is this correct that only the 2016 data is used for all you results? I think making this clear in a Data & Methods section would be nice.

Section 4.3, how is the two IEs calculated in this method? It seems you have retrievals (sample size ranging from 3 to 24) of cloud and corresponding aerosol properties for each cloudy period, from which IEs are derived? Making this clearer would be nice.

Check for spelling: P11 Line232: Ascension; Line 228: August or September? Fig. 5: daily.

P11 L220, what does "cloud inversion" mean?

Define abbreviations at first use: SNR, ATB

Fig. 7-9, validations are better illustrated with scatter plots with $R^2$ values provided, similarly as in Fig. 11.

Fig. 6, x-axis font needs to be adjusted.

Fig. 8-9, the range of y-axis needs to be adjusted to remove the empty space above and to see the variability in the variables better, and why filling in no-retrieval period with straight lines, if you don't have values for that time period?

Data doi needs to be provided.

References:

Adebiyi, A. A. and Zuidema, P.: The role of the southern African easterly jet in modifying the southeast Atlantic aerosol and cloud environments, Q. J. Roy. Meteor. Soc., 142, 1574–1589, https://doi.org/10.1002/qj.2765, 2016.

Adebiyi, A. A., Zuidema, P., and Abel, S. J.: The Convolution of Dynamics and Moisture with the Presence of Shortwave Absorb ing Aerosols over the Southeast Atlantic, J. Climate, 28, 1997–2024, https://doi.org/10.1175/JCLI-D-14-00352.1, 2015.

Zhang, J. and Zuidema, P.: The diurnal cycle of the smoky ma- rine boundary layer observed during August in the remote southeast Atlantic, Atmos. Chem. Phys., 19, 14493–14516, https://doi.org/10.5194/acp-19-14493-2019, 2019.

Zhang, J. and Zuidema, P.: Sunlight-absorbing aerosol amplifies the seasonal cycle in low-cloud fraction over the southeast Atlantic, Atmos. Chem. Phys., 21, 11179–11199, https://doi.org/10.5194/acp-21-11179-2021, 2021.

Zuidema, P., Sedlacek, A. J., Flynn, C., Springston, S., Delgadillo, R., Zhang, J., Aiken, A. C., Koontz, A., and Muradyan, P.: The Ascension Island Boundary Layer in the Remote Southeast At- lantic is Often Smoky, Geophys. Res. Lett., 45, 4456–4465, https://doi.org/10.1002/2017GL076926, 2018.

---

## Author Comment (AC1)

**Review #1 of "Aerosol first indirect effect of African smoke in marine stratocumulus clouds over Ascension Island, south Atlantic Ocean" by M. de Graaf et al.**

This study uses a single instrument to study aerosol first indirect effect with one month of data collected at the Ascension Island in the middle of the south Atlantic Ocean during the southern African biomass burning season. The manuscript is relatively well written, and the scope of the study is properly fitted for ACP, however, the current form of the manuscript reads too technical (more suited for AMT), and the significance of the results shown needs to be assessed before it can be considered for publication. Moreover, upon addressing the above mentioned points, the structure of the manuscript also needs to be revised for the ease of readership.

The reviewer is thanked for the thorough and careful review of the manuscript. Many issues were raised, and we have tried to answer them all satisfactorily below. The manuscript was completely revised, changing it from a technical reading to a geophysical article about the aerosol-cloud interactions over Ascension Island in the context of the stratocumulus to cumulus transition, which is important in this area. The suggestions and comments from the reviewer were very helpful to improve the manuscript to be more useful for the scientific community.

Below the comments are answered in detail. It is indicated when the manuscript was changed to comply to the raised issue.

**Main concerns:**

1. Statistical significances of the analyses (all 3 methods associated with Fig. 4-6) need to be included and discussed. Working with only one month of available can be quite challenging, but I believe one can still make valuable statements with proper significance assessments.
   Agreed. The statistical significance, comparative numbers from different other studies and a discussion on the meteorological conditions during the campaign, affecting the results, are now added to the manuscript.

2. attribution of the Twomey effect requires constant cloud macrophysical properties, e.g., cloud LWP, and environmental conditions, I understand this can be difficult with less than 40 sample size, but these limitations/assumptions need to be acknowledged when interpreting the results and making attributions (i.e., saying these indicate the Twomey effect).
   Agreed. A discussion section was added to discuss the limitations and the the manuscript was restructured to better describe the results and the uncertainties.

3. Section 5.2 confuses me, if one cannot validate the representativeness of

the retrieved cloud properties, how can one study the interactions of them with retrieved aerosol properties? Moreover, if the Radar retrievals of cloud properties is really problematic and biased (re retrieved using daily averaged or some assumed Nd values are certainly not suited for aerosol-cloud interactions studies), what's the point of validating Lidar Nd with Radar retrievals? Furthermore, it doesn't validate a retrieved variable when another variable retrieved using the same instrument is involved in the validation.

Radar can be used to estimate cloud droplet number density and effective radius using the well-known and often used methods described by Frisch *et al.* (1995) and Frisch *et al.* (2002). If only radar measurements are available, an estimate (assumption) on cloud number density can be used to determine effective radius. This is done in Frisch et al. (2002). However, an independent estimate of cloud droplet number density can be achieved using the lidar measurements. In this way, the complementary measurements should yield cloud parameters that are more accurate and less sensitive to assumptions. The section describing this in the manuscript was rewritten to show this more clearly.

4. The current form of the manuscript reads too technical, especially Sections 2.1-2.4 and 5.2. These technical details can be condensed and summarized in the main text, with details provided in an appendix or a supplement.
   The manuscript was completely restructured, following the suggestions of the reviewers. The theory was moved to the Appendix, along with technical sections. The introduction was rewritten, to include literature references.

5. The validation section needs to be moved up before showing the results. How can a reader interpret these results without knowing the retrievals that these statements based on are validated?
   The manuscript was restructured. First the campaign and the lidar measurement results are shown, with comparisons from other sources. Then the ACI are determined and described. The discussion section now shows the uncertainties that are associated with the lidar and radar cloud retrievals.

6. The current form of the Conclusions reads like a summary and repetitive of what have been stated. Emphasizing on the advantages (and caveats) and implications of the study would be very helpful.
   The conclusions (and introduction and results) sections have been completely rewritten.

**Minor comments:**

The authors tend to state existing knowledges without providing references, for example:

1. P1 Line17 and P8 Line166, the typical thermodynamical structure of the MBL clouds over Ascension during the dry season can be found in Zhang & Zuidema 2019 ACP. The reference was added.

2. P2 Line33 and P8 Line173, Zuidema et al. 2018 GRL provides a more updated overview on LASIC and some first results. The reference was added.

3. When providing information on the transport, seasonality, and distribution of the smoke aerosols to set up the context on the complex environment within which smoke-cloud interactions manifest over the SE Atlantic during the southern African biomass burning season, Adebiyi & Zuidema 2016 QJRMS, Adebiyi et al. 2015 JClimate, and Zhang & Zuidema 2021 are suitable references. The introduction was rewritten and several reference were added to properly describe context of smoke in the African biomass burning season.

4. P11 L222, reference for the theoretically feasible values? McComiskey *et al.* (2009). This was added.

5. P13 L284, "from the literature", which one? L273: A typical value for Nd for marine, low-level stratocumulus clouds is $100 \pm 70$ cm$^{-3}$ (Davidson et al., 1984; Martin et al., 1994). This was changed in the manuscript.

Since cloud properties retrieved by the lidar only represents cloud base values, and (as the authors also mentioned) cloud droplet size is highly dependent on height, I wonder if it's more appropriate to indicate that this study focuses on the aerosol indirect effect at cloud base in the title, with "... at cloud base of marine stratocumulus clouds ..."?
Fair enough. The title was changed, to reflect this and the concern by reviewer #2 about the cloud type.

The current introduction is too thin, introduction to existing knowledges on aerosol cloud interactions over the region is needed to set up the scientific question. The introduction to the Ascension Island and its environment, i.e., the smoky SE Atlantic during dry season, and the campaign info need to be moved up, preferrable to the introduction. When reading the current manuscript, a reader has no idea of the context (the condition under which these measurements were made) until P7 Section 3.
The introduction was rewritten

If the 2017 measurements were affected by alignment problems, why all your proof of concept exemplary figures show 27 Aug 2017? Why not use a day from 2016?
A day from 2016 is now used.

Why the sample sizes of the 3 methods not consistent? (37 in Fig. 4, 39 in Fig. 5, and 32 in Fig. 6). This needs to be justified.

The number of samples change, because for each method different criteria are used, as described in the manuscript. E.g. in the second method only clouds between -300m and +1000m from the cloud base are considered, whereas in method three only clouds are considered when a successful extinction profile was also retrieved from beneath the cloud. In the first method all successful cloud retrievals were used during the defined days. This is different between the methods, not necessarily inconsistent. However, care has to be taken when comparing different results, this is now better explained in the manuscript.

Is this correct that only the 2016 data is used for all you results? I think making this clear in a Data & Methods section would be nice.

Yes, only 2016 data was used to compute the IE. Validation was also done on 2017 data, due to the unavailability of radar data in 2016. This was made clearer in the manuscript.

Section 4.3, how is the two IEs calculated in this method? It seems you have retrievals (sample size ranging from 3 to 24) of cloud and corresponding aerosol properties for each cloudy period, from which IEs are derived? Making this clearer would be nice.

This is explained in the manuscript. Three to a maximum of 24 samples of 30 s intervals were averaged, from which the IEs (now ACI to be consistent with McComiskey *et al.*, 2009) are calculated. In Fig. 7, the sample size (time period length) is indicated by the color.

**Check for spelling:** P11 Line232: Ascension; Line 228: August or September? Fig. 5: daily.

Done

P11 L220, what does "cloud inversion" mean?

Lidar inversions, this was changed

Define abbreviations at first use: SNR, ATB

SNR was defined, ATB was not found

Fig. 7-9, validations are better illustrated with scatter plots with R2 values provided, similarly as in Fig. 11.

Done. Fig. 7 (now 3) includes a scatter plot of the measurements, and Fig 8 and 9 were revisited and now show temporal plots and scatter plots of the retrievals, including linear fits to the comparisons.

Fig. 6, x-axis font needs to be adjusted.

Done, the figure was updated and improved.

Fig. 8-9, the range of y-axis needs to be adjusted to remove the empty space above and to see the variability in the variables better, and why filling in no-retrieval period with straight lines, if you don't have values for that time period?

Done. The figures were updated, see above.

Data doi needs to be provided.

Done.

**References:**

- Adebiyi, A. A. and Zuidema, P.: The role of the southern African easterly jet in modifying the southeast Atlantic aerosol and cloud environments, Q. J. Roy. Meteor. Soc., 142, 1574–1589, https://doi.org/10.1002/qj.2765, 2016.

- Adebiyi, A. A., Zuidema, P., and Abel, S. J.: The Convolution of Dynamics and Moisture with the Presence of Shortwave Absorb ing Aerosols over the Southeast Atlantic, J. Climate, 28, 1997–2024, https://doi.org/10.1175/JCLI-D-14-00352.1, 2015.

- Zhang, J. and Zuidema, P.: The diurnal cycle of the smoky ma- rine boundary layer observed during August in the remote southeast Atlantic, Atmos. Chem. Phys., 19, 14493–14516, https://doi.org/10.5194/acp-19-14493-2019, 2019.

- Zhang, J. and Zuidema, P.: Sunlight-absorbing aerosol amplifies the seasonal cycle in low-cloud fraction over the southeast Atlantic, Atmos. Chem. Phys., 21, 11179–11199, https://doi.org/10.5194/acp-21-11179-2021, 2021.

- Zuidema, P., Sedlacek, A. J., Flynn, C., Springston, S., Delgadillo, R., Zhang, J., Aiken, A. C., Koontz, A., and Muradyan, P.: The Ascension Island Boundary Layer in the Remote Southeast At- lantic is Often Smoky, Geophys. Res. Lett., 45, 4456–4465, https://doi.org/10.1002/2017GL076926, 2018.

- Frisch, A. S., Fairall, C. W., and Snider, J. B.: Measurement of Stratus Cloud and Drizzle Parameters in ASTEX with a Ka-Band Doppler Radar and a Microwave Radiometer, Journal of the Atmospheric Sciences, 52, 2788–2799, https://doi.org/10.1175/1520- 0469(1995)052¡2788:MOSCAD¿2.0.CO;2, 1995.

- Frisch, A. S., Shupe, M., Djalalova, I., Feingold, G., and Poellot, M.: The Retrieval of Stratus Cloud Droplet Effective Radius with Cloud Radars, Journal of Atmospheric and Oceanic Technology, 19, 835–842, https://doi.org/10.1175/1520-0426(2002)019¡0835:TROSCD¿2.0.CO;2, 2002.

---

## Author Comment (AC2)

**Review #2 of "Aerosol first indirect effect of African smoke in marine stratocumulus clouds over Ascension Island, south Atlantic Ocean" by M. de Graaf et al.**

In this work, the authors report on cloud microphysical properties of low-level marine clouds inferred from UV-polarization lidar. The lidar was deployed during the dry season months of 2016 and 2017 on a remote south Atlantic island. A new technique developed in an earlier work (Donovan et al., 2015) was applied to infer microphysical parameters (aerosol optical depth, cloud droplet effective radius, and cloud droplet number concentration) and compared with in situ measurements from AERONET and instruments deployed during the ARM LASIC campaign. Although the work provides valuable insights into the complex ACI at Ascension Island, the authors have contributed some preliminary understanding to processes contributing to the observed interactions due to smoke intrusions into the cloud deck, environmental and instrumental effects on measured uncertainties, but they do not relate their findings to the growing body of literature in this region for comparison. I believe this paper is worthy of publication after these components have been more clearly addressed for compliance with ACP criteria, therefore major revision is recommended.

*The reviewer is thanked for the extensive and helpful review of the paper. We agree that the paper should be related to large body of existing studies. The reviewer is acknowledged for the many suggestions for accompanying papers, which are now cited in the new manuscript. The manuscript was completely rewritten, especially the introduction and conclusion sections, to reflect the contributions to this field.*

*Below, the suggestions and comments raised by the reviewer are all answered in detail, and the changes to the manuscript are indicated.*

**Major comments:**

This paper would benefit from a more complete description of the context of the work and its motivation. To this end, the introduction should be expanded. Particularly, the authors provide no description of the first indirect / Twomey effect in the introduction and only offer a vague claim that drizzle accompanying low-level marine clouds can be modulated by an interaction with aerosol. Many modeling and observational studies have conducted examinations of aerosol effects on low-level marine clouds (e.g. McComiskey et al. (2009), Yamaguchi et al. (2017)), and this work should be explicitly placed in that context. Specific focus on absorbing aerosol, such as the biomass burning smoke that impact the cloud deck that reaches Ascension Island has also been investigated (e.g. Ajoku et al. (2021), Diamond et al. (2018), Kacarab et al. (2020), Painemal et al. (2014)). These and related works should be cited to give context for the aerosol expected to drive changes in the Ascension Island microphysics and the potential environmental, compositional, and physical factors contributing to these changes. The authors should also describe what makes UV-polarization lidar advantageous over other commonly applied methods as well as its limitations.

The introduction has been completely rewritten, following the recommendations of the reviewer. The recommended works have all been added and cited to provide a proper context of the paper, for which the reviewer is thanked. The advantage of limitations of both the lidar and the radar methods used are described in more detail, a discussion section was added in which the results are compared with existing studies on retrieval of cloud parameters.

- The authors should consider restructuring the paper's outline of sections, namely the order of the theory, measurements, and methods, as these sections appear to be interspersed throughout the paper rather than contained within specifically focused sections. It would benefit understanding and context of the work if "Section 3: Measurement" campaign was placed before "Section 2: Theory" as some of the discussion in Section 2 references data described in Section 3 (Fig. 1).
  The paper has been restructured following the recommendation of both reviewers: The measurement campaign section now follows the introduction, and the theory section was moved to the appendix. The paper now clearly describes the aerosol-cloud interactions using a variety of methods, all with their merits and drawbacks. The technical description of the measurements was moved from the main story.

- "Section 5.3 Cloud Base Height validation" does not report on any aerosol-cloud interaction results and only gives a comparison between lidar-estimated cloud base height and two external estimates of cloud base. For this reason, it may be appropriate for this section to be moved to the supplement.
  The section was moved to the Appendix. We feel this is an important aspect of the assumptions used in the paper. In order to show the robustness of the method, the assumption of a proper cloud base height is essential. However, it was removed from the main story.

- More detail about the UV-polarization lidar used in this work should be provided. Specific details about the instrument itself, measurement frequency, uncertainties, and calibration should be included before introducing the theory equations in Section 2. Are the main results shown as daily averages? Were specific filtering techniques applied during averaging?
  The description of the lidar and measurements have been expanded. Calibration details and other details that are necessary for reproducing the results, but not necessary for interpretation of the results, are cited.
  The main results are differentiated into three main cases, clearly described in the manuscript. Some are daily averages, some are cases by case from selected cloudy intervals.
  The filtering techniques are described in the manuscript and the cited theses.

- As the authors have stated, it is customary to examine aerosol indirect effects by controlling for macrophysical (McComiskey et al., 2009; M. Miller et al., 2022) or meteorological (Scott et al., 2020) This was not done in

this work. The authors should speak more to how a lack of factor control on these measurements may impact the interpretation of the results.
Agreed. A section about the meteorological conditions during the campaign was added. A sidcussion of the impact of the meteorological conditions and its impact on the ACI results was added.

- Ascension Island lies at the terminating stage of the Southeast Atlantic stratocumulus-to-cumulus transition in the quiescent trade wind cumulus region. Zhang and Zuidema (2019) reported that the cloud types at Ascension are predominantly cumulus clouds with little vertical extent or cumulus clouds overlain by stratocumulus (two-layers), with single stratocumulus contributing less than 3% during the smoky season (August 2016 & 2017). The authors should describe how the specific cloud scenes were selected for the measurement comparisons and note, as in the title, that stratocumulus were the predominant cloud types observed and analyzed.
Agreed. The clouds that were selected were actually broken clouds over Ascension, so more likely cumulus clouds instead of stratocumulus. A paragraph on the measurement selection was added, showing a measurement sample with various cloud conditions and how the cloud selection was performed. The introduction has been changed to describe the paper in the context of the SCT and the title was changed.

- A broadened discussion comparing the retrieved microphysical parameters and computed aerosol indirect effects is necessary to provide more scientific basis to the report and interpretation of results. The authors should aim to answer specific questions about these results and their relation to measurements from other studies in relevant and related environments. How do the cloud droplet number and size inferred from this lidar technique compare to these parameters in other open ocean environments that are clean and impacted by smoke aerosol? The relative magnitude of the droplet number change appears to be much larger than that of the size change. Was this expected and consistent with previous work? If not, why? Additionally, the clean effective radius appears to be much smaller than the global average for warm clouds ($\tilde{1}4\ \mu m$). Can the authors ascribe this low value to a property of the observed clouds or environment? How do the computed indirect effects compare to other regionally and globally estimated aerosol indirect effects? Are the magnitudes of these results consistent with other pristine environments perturbed by strong pollution signals?
The introduction, discussion and conclusion sections were rewritten to describe the measurements in the context of the SCT. The $ACI_r$ found from the lidar measurements are at the high end of the ranges found in other papers, but consistent. The effective radius from the lidar is much smaller than the global average. This is mainly due to the sensitivity of the lidar to the cloud base. A comparison with radar retrievals at the cloud base

are consistent within the measurement uncertainty and comparative with previous lidar, in-situ and radar retrievals of the cloud droplet effective radius, if the strong dependence on height is taken into account. This is now discussed more extensively.
The indirect effects are at the high end of magnitudes found in other environments, but consistent with strong pollution events.

**Minor comments:**

The reader would benefit from having the aerosol indirect effect slopes summarized in the abstract.
Done

- Several cited papers in the main text are missing from the list of references, including: Bennartz (2007), Albrecht et al. (1998), Paluch et al. (1991). Done

- Line 39: Please provide a definition of "SNR" prior to using the acronym. Done

- Eq (3): what is $r_{atm}$? Is this supposed to be $r_{air}$ as in Eq(4)? Please be consistent with these variable names. Yes, done

- Eq(4): Based on the units of $r_{dryair}$ (J kg-1 K-1) and the fact that this equation is solving for the atmospheric density using ideal gas law, I believe this variable should be $R_{dryair}$, i.e. the universal gas constant for dry air, not the gas density of dry air. Correct, changed

- Line 87-89: What did the tests in which $S_{marine}$ and $S_{dark}$ were varied reveal about the sensitivity of the lidar ratio choices used in this work? A five percent change in AOT was found for changes in the lidar ratio within reasonable values. This was added to the manuscript.

- Figure 1: There is a discrepancy between the title label of this plot and the caption: the title shows 20170826, but the caption reads 27 Aug. 2017. Is there a reason for this discrepancy? It's 26 Aug. This was changed.

- Line 131: Please clarify the name of $\Gamma_l$. Is this an adiabatic lapse rate? Yes, this was added.

- Figure 3,4,5: Do these figures use data from both years or has 2017 data been excluded? Please clarify.

  Yes, only 2016 data were used to determine aerosol-cloud interactions. The section now opens with this statement.

- Line 167-169: Boundary layer and free tropospheric aerosol composition during the dry monsoonal season in the Southeast Atlantic has been characterized in previous work and should be cited (see (Dang et al., 2022; R.

Miller et al., 2021; Swap et al., 1996)).
The introduction and the measurement campaign section were rewritten and the references added.

- Line 193-194: How were the atmospheric layers (850 – 2150 m and 2150 – 5000 m) selected. Was the lidar backscatter or radiosonde profiles used to distinguish between cloud base – top and free troposphere?
  The lidar range was used to determine the altitude. The backscatter lidar quickviews were investigated by eye to determine a rough estimate of the vertical layers. An example of the quickview and selection process was added to the manuscript.

- Line 193-196: The authors should use consistent terminology when referring to the above-cloud atmospheric layer as either the "free troposphere" (as in Line 194) or "upper air" (as in Line 196).
  Done

- Line 205 – 207: The authors state: "It is assumed that aerosols between these levels have a significant impact on cloud forming." This statement is a bit vague and should provide evidence as to why it is believed that aerosol at these levels are most significant for cloud formation in this region.
  This was rephrased to state that the cloud base is the lidar-sensitive region, and the aerosol are sampled in this region as well.

- Line 225-227: The statement about "other meteorological conditions" contributing to retrievals with large numbers and uncertainties is vague. Can the authors point to specific meteorological conditions relevant to Ascension Island and the Southeast Atlantic Ocean that would contribute to such results? I would expect that meteorological conditions are fairly persistent and unchanged at this tropical site. Have the authors fully exhausted their assessment of uncertainty in the retrievals that could potentially lead to large numbers or uncertainty not explained by the meteorology?
  The meteorological conditions were checked by inspection of backtrajectories during the campaign and afterwards in the analyses, showing stable MBL conditions and variable upper air transport. In the manuscript a discussion is added citing new recent references describing the meteorological and climatological circumstances during the various measurement campaigns.

- Line 229-231: Shouldn't the months of discussion be September not August if referencing Figs. 6,7? The text states August in these passages.
  Yes, this was corrected.

- Line 229-230:What is the meaning of a "saturated Twomey effect"?
  This statement was deleted. No change is observed if all aerosols are activated.

- Line 230-231: The authors state they observe "elevated AOT" in Sept. 12-15 leading to near zero indirect effect (cloud drop number). This is a bit difficult to glean from Fig. 7 given that near zero indirect effect (cloud drop number) is observed for Sept. 9-10, which also had low AOT. Is this AOT elevation relative to the month observed, and what is the magnitude of this "elevation" relative to the seasonal or annual average in AOT? Zuidema et al. (2018) report on the boundary smoke aerosol loading during these periods, which may help the authors attain insight into the aerosol impact on the observed indirect effects.
  A new paragraph was introduced to discuss the AOT and aerosol concentrations during the campaign in relationship with climatological means. The AOT values are high due to smoke incursions, as described by Ryoo et al. (2022), but not extremely high values compared to August 2016 values, as described by Zuidema et al. (2018). This is now described more clearly in the manuscript.

- Line 234: please clarify "various parameters and instrument noise".
  This line was removed

- Line 234-235: Although a reference is provided for the 2017 indirect results being inconclusive, please provide a brief summary of how these results lead to an "inconclusive effect." In the context of the computed indirect effects, what does inconclusive mean?
  In 2016 the lidar was just been serviced by Leosphere which made that the alignment was better than in 2017 and thus the SNR was higher in 2016 than in 2017. Therefore, retrieval error in 2016 was 19.75% and in 2017 39.05%, due to the calibration, retrieval and measurement errors and the 2017 results provide no statistical significant ACI due to the large uncertainties. This referenced statement has now been added to the manuscript.

- Line 250: Can the authors provide a statistical significance value for the AOT vs AERONET correlation coefficient of 0.76?
  Pearson's statistical correlation coefficient was 0.76, showing strong correlation. The figure was changed to include a scatterplot of the measurements, and a linear fit was drawn to show the relation.

- Line 284-285: What is a typical cloud droplet size estimate and range for marine low-level clouds? Are these typical values consistent with having large cloud drop concentrations as observed in this study?
  Typical numbers range from a few microns at the cloud base to several tens of microns at cloud top for well developed clouds. Numbers from

various references have now been added to the manuscript and the results are discussed in the light of previous studies.

- Line 285-290: How was the Reff100 derived using LWP measurements from the MWR? The authors note, MWR-retrieved Reff100 was much more wildly varying than the lidar and cloud radars followed by a reference to Fig. 8, however, a comparison of lidar, cloud radar, and MWR retrievals is not shown. Why have the authors not shown the MWR-retrieved Reff100?

  The $R_{\text{eff}}^{100}$ was derived following a method described by Frisch et al. 2002. A MWR-derived $R_{\text{eff}}^{100}$ is discussed in one of the theses which are the basis for this paper, but the results ($R_{\text{eff}}^{100} > 15$ $\mu$m) were strongly inconsistent with non-drizzling clouds. The reason for this is unclear, but may point to biases in the LWP data used or an error in the implementation. Therefore the results are not shown in the paper. This statement was added to the manuscript.

- Line 293: Are the authors referring to liquid water path or the cloud droplet number density when it is stated that "this parameter was more than 5 times higher than the assumed 100 g m-2"? I assume this is the cloud droplet number concentration and the units should be cm-3.
  Yes, it should be cloud droplet number density. This and the unit were changed.

- Line 296-297: Zhang et al. (2011) is later referenced as a citation for the statement that cloud radii are strongly dependent on height in the cloud (Line 303-304). Please consolidate these statements or provide the citation the first time the statement is mentioned.
  Done

- Line 299-300: Please provide a citation describing higher radar measurement sensitivity to drizzle than lidar measurements.
  Several references have been added.

- Line 307-308:Please provide the correlation and statistical significance of the CBH correlation in these lines of text.
  Done

- Line 318-321: These lines do not contribute to a summary of the results of the paper and instead provide theory of the measurements used in this work. It is recommended that this material be moved to the theory section (Section 2).
  Done

- Line 334: Based on the results previously described, the indirect effect for cloud droplet effective radius should be negative, i.e. -0.18 ±06 $\mu$m, not positive.
  Corrected

- Data availability statement: Can the authors please provide a source to locate the freely available lidar data?
  Yes, done.

- Figure 6: Please extend the ticks of the x-axis and labels in both panels so that the dates can be clearly read. The numbers following the 10th of September are difficult to distinguish.
  The figure was updated and improved.

- Figure 7: There is a discrepancy between terminology in the figure and caption. The y-label shows AOD, while the caption references Aerosol Optical Thickness and AOT. Please choose a consistent terminology.
  Done. All changed into AOT.

- Figure 10: Can the authors provide the elevation of the main ARM site and airport site in the caption?
  Yes, done.

- Figure 11: Is the dashed line in this figure the 1:1 line or the regression? Please clarify.
  It's the 1:1 line, added.

- Where the authors have discussed or shown time series between measurements (lidar vs. radar, AERONET vs lidar, MWR vs lidar / radar), comparison plots (e.g. Figure 11) should also be provided with acknowledgement of the slope or bias in these comparisons.
  Done, all those figures have been revisited and now contain scatter plots with linear fits, showing slopes and biases.

**Technical Corrections**

All of the following technical correction were implemented as suggested:

- Line 48: Please correct "devided" to "divided".

- Line 202: Please correct "garantueed" to "guaranteed".

- Line 206: Please correct "forming" to "formation".

- Figure 5 caption: Please correct "daioly" to "daily".

**References**

1. Ajoku, O., Miller, A., & Norris, J. (2021). Impacts of aerosols produced by biomass burning on the stratocumulus-to-cumulus transition in the equatorial Atlantic. Atmospheric Science Letters, 22(4). Article.

2. Dang, C., Segal-Rozenhaimer, M., Che, H., Zhang, L., Formenti, P., Taylor, J., et al. (2022). Biomass burning and marine aerosol processing over the southeast Atlantic Ocean: a TEM single-particle analysis. Atmospheric Chemistry and Physics, 22(14), 9389-9412. Article.

3. Diamond, M., Dobracki, A., Freitag, S., Griswold, J., Heikkila, A., Howell, S., et al. (2018). Time-dependent entrainment of smoke presents an observational challenge for assessing aerosol-cloud interactions over the southeast Atlantic Ocean. Atmospheric Chemistry and Physics, 18(19), 14623-14636. Article.

4. Kacarab, M., Thornhill, K., Dobracki, A., Howell, S., O'Brien, J., Freitag, S., et al. (2020). Biomass burning aerosol as a modulator of the droplet number in the southeast Atlantic region. Atmospheric Chemistry and Physics, 20(5), 3029-3040. Article.

5. McComiskey, A., Feingold, G., Frisch, A., Turner, D., Miller, M., Chiu, J., et al. (2009). An assessment of aerosol-cloud interactions in marine stratus clouds based on surface remote sensing. Journal of Geophysical Research-Atmospheres, 114. Article.

6. Miller, M., Mages, Z., Zheng, Q., Trabachino, L., Russell, L., Shilling, J., & Zawadowicz, M. (2022). Observed Relationships Between Cloud Droplet Effective Radius and Biogenic Gas Concentrations in Summertime Marine Stratocumulus Over the Eastern North Atlantic. Earth and Space Science, 9(2). Article.

7. Miller, R., McFarquhar, G., Rauber, R., O'Brien, J., Gupta, S., Segal-Rozenhaimer, M., et al. (2021). Observations of supermicron-sized aerosols originating from biomass burning in southern Central Africa. Atmospheric Chemistry and Physics, 21(19), 14815-14831. Article.

8. Painemal, D., Kato, S., & Minnis, P. (2014). Boundary layer regulation in the southeast Atlantic cloud microphysics during the biomass burning season as seen by the A-train satellite constellation. Journal of Geophysical Research-Atmospheres, 119(19), 11288-11302. Article.

9. Scott, R., Myers, T., Norris, J., Zelinka, M., Klein, S., Sun, M., & Doelling, D. (2020). Observed Sensitivity of Low-Cloud Radiative Effects to Meteorological Perturbations over the Global Oceans. Journal of Climate, 33(18), 7717-7734. Article.

10. Swap, R., Garstang, M., Macko, S., Tyson, P., Maenhaut, W., Artaxo, P., et al. (1996). The long-range transport of southern African aerosols the tropical South Atlantic. Journal of Geophysical Research-Atmospheres, 101(D19), 23777-23791. Article.

11. Yamaguchi, T., Feingold, G., & Kazil, J. (2017). Stratocumulus to Cumulus Transition by Drizzle. Journal of Advances in Modeling Earth Systems, 9(6), 2333-2349. Article.

12. Zhang, J., & Zuidema, P. (2019). The diurnal cycle of the smoky marine boundary layer observed during August in the remote southeast Atlantic. Atmospheric Chemistry and Physics, 19(23), 14493-14516. Article.

13. Zuidema, P., Sedlacek, A., Flynn, C., Springston, S., Delgadillo, R., Zhang, J., et al. (2018). The Ascension Island Boundary Layer in the Remote Southeast Atlantic is Often Smoky. Geophysical Research Letters, 45(9), 4456-4465. Article. Citation: https://doi.org/10.5194/acp-2022-473-RC2

---

## Referee Report (RR1)

**Review of "Aerosol first indirect effect of African smoke in marine stratocumulus clouds over Ascension Island, south Atlantic Ocean" by M. de Graaf et al.**

I appreciate the efforts the authors put into revising the manuscript, which has been much improved compared to the original version. Most of my concerns/comments are well addressed.

That said, after reading the current version of the manuscript, I do have some remaining points that I would like the authors to consider first before finalizing for publication.

I recommend publication after minor revisions.

**Main comments:**

1. Both in the abstract and in the conclusion (where readers read the most), the authors point out their estimates of $ACI_N$ and $ACI_r$ (line 7 and 294). These ACI metrics are supposed to be unitless as these metrics are calculated as ln-ln regressions. Furthermore, please double check the sign convention of your formula for the 2 ACI metrics (Eq. 1 and 2). The sign convention is supposed to make these ACI metrics appear as positive values, such that decreasing effective radius with increasing aerosol is indicated by a positive value (by a -ve sign infront), and increasing Nd with increasing aerosol is also indicated by a positive value (without the -ve sign infront).
    a. Your Fig. 7 confused me, according to your Eq. 2, panel a) should have a negative ACI value, and panel b) should have a positive value according to your Eq. 1.
    b. Your Eq. 2 is inconsistent with McComiskey et al. (2009).

2. I appreciate the fact that a comprehensive overview of various aerosol effects on marine warm clouds is now included in the introduction. That said, I do notice that these references are mostly rather old studies (they are nice studies and should be referenced here). What I recommend is to include some newer studies (in addition to the ones already cited), especially those coming out of ORACLES/CLARIFY/LASIC campaigns that took place between 2016-2018, to show what we have learned so far thanks to all these amazing campaigns. For instance…
    a. Zhang & Zuidema (2021) found that the changing smoke vertical distribution during the dry season over the remote SEA (Ascension Island) leads to different cloud adjustments, and thereby an amplified low-cloud fraction seasonal cycle is observed in the presence of smoke.
    b. Diamond et al. (2022) use a combination of regional and high-resolution modeling to show large-scale smoke–circulation interactions strongly modules the SCT in this region, which has been overlooked previously.
    c. Gupta et al. (2022) report the same ACI metrics as in this study but based on ORACLES airborne measurements.
    I think all these newer papers fit nicely to your discussion in the paragraph of lines 36-49.

3. Regarding Fig. 8. How is the uncertainty bar for each ACI estimate quantified? A sentence clarifying this would be nice, so that the readers will have an idea of what's going on for those estimates with huge uncertainty bars.

4. Regarding the fact that you have many ACI estimates outside the "theoretical" bounds suggested by McComiskey et al. (2009), I want to say that I don't think your values are unphysical or unrealistic, besides what you already stated in the text, I want to add that:
    a. For $|ACI_N| > 1$, a bound of 1 only makes sense to me when aerosol number is used (so that a value of 1 indicates total activation of aerosol particles into cloud droplets). The fact that you are using mean extinction coefficient below clouds may lead to values larger than 1.
    b. For $|ACI_r| > 0.33$, a bound of 0.33 is true only when LWP is controlled when calculating $ACI_r$, as indicated in McComiskey et al. (2009) Eq. 1b. The fact that you are not controlling LWP when calculating $ACI_r$ may lead to values larger than 0.33 due to covarying LWP.

**Minor comments:**

Line 93, this sentence suggests that the ARM site is at 859m above sea level, please double check and revise.

Line 165-166, how do you define periods of clear sky and cloudy sky, based on what metric(s)? How many periods of clear sky and cloudy sky are there during the studied period?

Line 172-174 and Fig. 6, I am confused about the #6 and #31 under 'clean' and 'mixed' in Fig. 6's caption. I know there are 5 clean days and 8 mixed days according to Fig. 5, so they must indicate the number of cloudy periods of each category, i.e., 6 cloudy periods in clean days and 31 cloudy periods in mixed days, correct? Then in the text you say 6 and 31 indicate the number of cloud free periods and cloudy periods. Please clarify.

Caption of Fig. 10, inconsistent with your legend, where you indicate variance is shown in black and error in red.

Line 254, "Nd" -> "$N_d$"

Line 255 and 258, unit should be cm^-3

Line 267, "… correlation was found…" what correlation? Word missing.

Line 395, "product" -> "produce"

**References:**

Zhang, J. and Zuidema, P.: Sunlight-absorbing aerosol amplifies the seasonal cycle in low-cloud fraction over the southeast Atlantic, Atmos. Chem. Phys., 21, 11179–11199, https://doi.org/10.5194/acp-21-11179-2021, 2021.

Diamond, M. S., Saide, P. E., Zuidema, P., Ackerman, A. S., Doherty, S. J., Fridlind, A. M., Gordon, H., Howes, C., Kazil, J., Yamaguchi, T., Zhang, J., Feingold, G., and Wood, R.: Cloud

adjustments from large-scale smoke–circulation interactions strongly modulate the southeastern Atlantic stratocumulus-to-cumulus transition, Atmos. Chem. Phys., 22, 12113–12151, https://doi.org/10.5194/acp-22-12113-2022, 2022.

Gupta, S., McFarquhar, G. M., O'Brien, J. R., Poellot, M. R., Delene, D. J., Chang, I., Gao, L., Xu, F., and Redemann, J.: In situ and satellite-based estimates of cloud properties and aerosol–cloud interactions over the southeast Atlantic Ocean, Atmos. Chem. Phys., 22, 12923–12943, https://doi.org/10.5194/acp-22-12923-2022, 2022.

---

## Author Response (AR2)

**Review of 'Aerosol first indirect effect of African smoke at cloud base of marine cumulus clouds over Ascension Island, south Atlantic Ocean'**

I appreciate the efforts the authors put into revising the manuscript, which has been much improved compared to the original version. Most of my concerns/comments are well addressed.

That said, after reading the current version of the manuscript, I do have some remaining points that I would like the authors to consider first before finalizing for publication.

The reviewer is thanked for the time and effort spent on the continued review of the manuscript, which has made it much stronger. This is highly appreciated. Below, the remaining concerns are addressed and the changes to the manuscript are indicated.

I recommend publication after minor revisions.

Main comments:

1. Both in the abstract and in the conclusion (where readers read the most), the authors point out their estimates of ACIN and ACIr (line 7 and 294). These ACI metrics are supposed to be unitless as these metrics are calculated as ln-ln regressions. Furthermore, please double check the sign convention of your formula for the 2 ACI metrics (Eq. 1 and 2). The sign convention is supposed to make these ACI metrics appear as positive values, such that decreasing effective radius with increasing aerosol is indicated by a positive value (by a -ve sign infront), and increasing Nd with increasing aerosol is also indicated by a positive value (without the -ve sign infront).

   Thanks for the clarification, indeed the metrics were not consistent with McComiskey et al (2009). This has now been corrected. The sign in Eq. 2 was changed, the ACI numbers are dimensionless in the manuscript and the figures and numbers are now in agreement with the above changes.

   - Your Fig. 7 confused me, according to your Eq. 2, panel a) should have a negative ACI value, and panel b) should have a positive value according to your Eq. 1.
     Corrected

   - Your Eq. 2 is inconsistent with McComiskey et al. (2009)."
     Corrected

2. I appreciate the fact that a comprehensive overview of various aerosol effects on marine warm clouds is now included in the introduction. That said, I do notice that these references are mostly rather old studies (they

are nice studies and should be referenced here). What I recommend is to include some newer studies (in addition to the ones already cited), especially those coming out of ORACLES/CLARIFY/LASIC campaigns that took place between 2016- 2018, to show what we have learned so far thanks to all these amazing campaigns. For instance...

- Zhang & Zuidema (2021) found that the changing smoke vertical distribution during the dry season over the remote SEA (Ascension Island) leads to different cloud adjustments, and thereby an amplified low-cloud fraction seasonal cycle is observed in the presence of smoke.
- Diamond et al. (2022) use a combination of regional and high-resolution modeling to show large-scale smoke–circulation interactions strongly modules the SCT in this region, which has been overlooked previously.
- Gupta et al. (2022) report the same ACI metrics as in this study but based on ORACLES airborne measurements.

I think all these newer papers fit nicely to your discussion in the paragraph of lines 36-49.

The reviewer is thanked for all the references, which has made the paper much more relevant. The first two of the suggestions above have been added to the introduction section, the third to the relevant section on ACI's.

3. Regarding Fig. 8. How is the uncertainty bar for each ACI estimate quantified? A sentence clarifying this would be nice, so that the readers will have an idea of what's going on for those estimates with huge uncertainty bars.

The errors bars are the standard deviation of the measurements during each selected interval. A sentence explaining the error bars is in the caption of the figure.

4. Regarding the fact that you have many ACI estimates outside the "theoretical" bounds suggested by McComiskey et al. (2009), I want to say that I don't think your values are unphysical or unrealistic, besides what you already stated in the text, I want to add that:

a. For $|\mathrm{ACI}_N| > 1$, a bound of 1 only makes sense to me when aerosol number is used (so that a value of 1 indicates total activation of aerosol particles into cloud droplets). The fact that you are using mean extinction coefficient below clouds may lead to values larger than 1.

b. For $|\mathrm{CI}_r| > 0.33$, a bound of 0.33 is true only when LWP is controlled when calculating $\mathrm{ACI}_r$, as indicated in McComiskey et al. (2009) Eq. 1b. The fact that you are not controlling LWP when calculating $\mathrm{ACI}_r$ may lead to values larger than 0.33 due to covarying LWP.

The discussion on the ACI's was adapted to reflect these considerations.

Minor comments:

Line 93, this sentence suggests that the ARM site is at 859m above sea level, please double check and revise.
The text was changed to make the altitudes of the locations and the volcanic peak more clear.

Line 165-166, how do you define periods of clear sky and cloudy sky, based on what metric(s)? How many periods of clear sky and cloudy sky are there during the studied period?
The clear and cloudy parts of the measurements were selected by visual inspection of the lidar quicklooks. The number of clear sky periods was 29 in 2016, and the number of cloudy sky periods was 43 in 2016. Details are published in Tenner et al, 2017. An explanatory sentence was added to the manuscript.

Line 172-174 and Fig. 6, I am confused about the # 6 and # 31 under 'clean' and 'mixed' in Fig. 6's caption. I know there are 5 clean days and 8 mixed days according to Fig. 5, so they must indicate the number of cloudy periods of each category, i.e., 6 cloudy periods in clean days and 31 cloudy periods in mixed days, correct? Then in the text you say 6 and 31 indicate the number of cloud free periods and cloudy periods. Please clarify.
This was a typo, confusing 'clean' and 'clear' The number of 6 and 31 both refer to cloudy periods, one during clear days (low smoke concentration) and one during mixed days (high smoke concentration). The text has been changed to correctly state the number of clean and mixed days, and clear and cloudy periods

Caption of Fig. 10, inconsistent with your legend, where you indicate variance is shown in black and error in red.
Correct. This was changed.

Line 254, "Nd" $->$ "N$_d$"
Corrected

Line 255 and 258, unit should be cm$^-$3
Corrected

Line 267, "... correlation was found..." what correlation? Word missing.
"No" correlation was found. Corrected.

Line 395, "product" $->$ "produce"
Corrected.